# Wyckoff Transformer: Generation of Symmetric Crystals

## Abstract

We propose Wyckoff Transformer, a generative model for materials conditioned on space group symmetry. Most real–world inorganic materials have internal symmetry beyond lattice translation. Symmetry rules that atoms obey play a fundamental role in determining the physical, chemical, and electronic properties of crystals. These symmetries determine stability, and influence key material structural and functional properties such as electrical and thermal conductivity, optical and polarization behavior, and mechanical strength. And yet, despite the recent advancements, state–of–the–art diffusion models struggle to generate highly symmetric crystals. We use Wyckoff positions as the basis for an elegant, compressed, and discrete structure representation. To model the distribution we develop a permutation–invariant autoregressive model based on Transformer and absence of positional encoding. Our experiments demonstrate that Wyckoff Transformer has the best performance in generating novel diverse stable structures conditioned on the symmetry space group, while also having competitive metric values when compared to model not conditioned on symmetry. We also show that it is competitive in predicting formation energy, band gap, mechanical properties, and thermal conductivity.

## 1 Introduction

Space of all possible combinations of atoms forming periodic structures is intractably large. It is not possible to screen it fully, even with a fast machine learning algorithm. Practical materials, however, occupy only a small part of it. Firstly, they must correspond to an energy minimum. Secondly, occupying an energy minimum is not sufficient to establish if the material is synthesizable or indeed experimentally stable. Having a generative model that outputs a priori stable materials is a step towards speeding up automated material design by orders of magnitude.

### 1.1 Space groups and Wyckoff positions

A crystal structure can be systematically described through its lattice and atomic basis. The lattice provides a repeating geometric framework, defined as an infinite periodic arrangement of points in space. Based on interactions between the constituent electrons and nuclei, atoms rearrange into such a lattice and, therefore, follow a finite set of symmetries: the group of all such symmetry operations that uniquely define the periodic arrangement is called the space group of the crystal. These arrangements in a crystal are governed by a finite set of symmetry operations, such as rotations, reflections, inversions, and translations. These operations combine to form the 230 distinct space groups, which serve as a comprehensive classification system for all possible crystal symmetries in three dimensions. Each space group defines the unique symmetry properties of a crystal structure, defining the allowable positions for atoms within the unit cell. This ensures that every crystal possesses at least the simplest level of symmetry, referred to as P1 symmetry, which involves only translational symmetry. The atomic basis specifies the arrangement of atoms associated with each lattice point, thus defining the overall crystal structure.

Importantly, most known crystals have internal symmetry, see figure 1. Those symmetries are not merely a mathematical observation; optical, electrical, magnetic, structural and other properties are determined by symmetry, as shown by Malgrange et al. (2014); Yang et al. (2005), as well as our results in section 3.2.

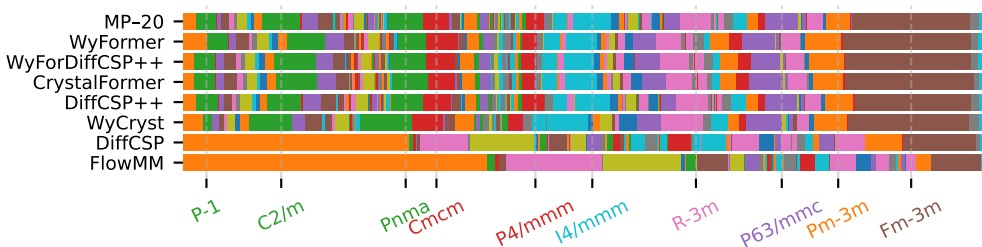

Figure 1: Distribution of space groups in MP–20 dataset Xie et al. (2021) and the generated samples. 10 space groups most frequent in MP–20 are labeled, 98% of MP–20 structures belong to symmetry groups other that P1. Plot design by Levy et al. (2024). The comparison of the distribution of generated samples' space groups to the original sample distribution is present in Table 2, column Space Group $\chi^2$.

Within a given space group, a subgroup forms the site symmetry, referring to the set of symmetry operations that leave a specific point in the crystal invariant. These operations describe the local symmetrical environment, such as mirror, screw axis, or inversions centered on that a given region. Higher site symmetry is in regions where multiple symmetry elements intersect, while those with lower site symmetry include only one symmetry operation. Taking space group 225 Fm-3m as an example, site symmetry subgroup m-3m represents a highly symmetric environment like the center of a cubic unit cell, where multiple symmetry elements intersect, including mirror planes and a 3–fold rotoinversion axis. In contrast, another lower site symmetry subgroup .3m corresponds to a less symmetric environment with only a 3–fold rotation axis and a mirror plane.

These site symmetry points, classified by their symmetry properties, are grouped into Wyckoff positions (WPs) (Wyckoff, 1922). Mathematically, a WP encompasses all points whose site symmetry groups are conjugate subgroups of the full space group Kantorovich (2004). Each WP is characterized by two key attributes:

1. Site symmetry

2. Symmetry equivalence: two different Wyckoff positions in the same space group can share the same site symmetry but may still be symmetry equivalent. This equivalence arises when the Wyckoff positions can be mapped onto each other using higher–order symmetry operations, such as those defined by the Euclidean normalizer of the space group. These symmetry–equivalent WPs form the basis for enumeration and augmentation in the subsequent sections of this work.

WPs for a given space group are commonly enumerated by Latin letters in the order of multiplicity, the number of equivalent atomic positions in a crystal structure that are related by the symmetry operations of the space group. WPs are denoted by a combination of the multiplicity value and the letter, e. g. `2a`. The number of distinct WPs in a space group is finite, ranging from a single WP in the simplest symmetry group P1 to as many as 27 in the most complex space groups. These classifications enable the description of not only discrete points but also more complex geometric features. For example, some Wyckoff positions represent 1D lines, 2D planes, or even open 3D regions within the unit cell, depending on the symmetry constraints. This flexibility underscores the utility of Wyckoff positions in describing diverse crystallographic arrangements. By introducing these fundamental concepts – lattice, atomic basis, space groups, site symmetry, and Wyckoff positions – this framework provides a foundation for understanding crystal structures. See also Appendix A for an illustration.

### 1.2 OUR CONTRIBUTION

Our contribution can be summarized as follows:

1. Representing a crystal as an unordered set of tokens fused from the chemical element and Wyckoff position; section 2.1.

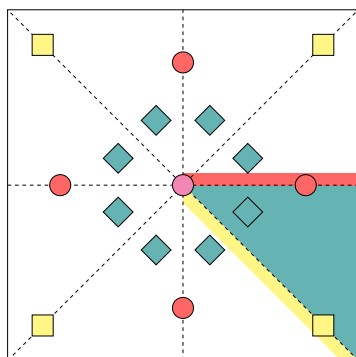

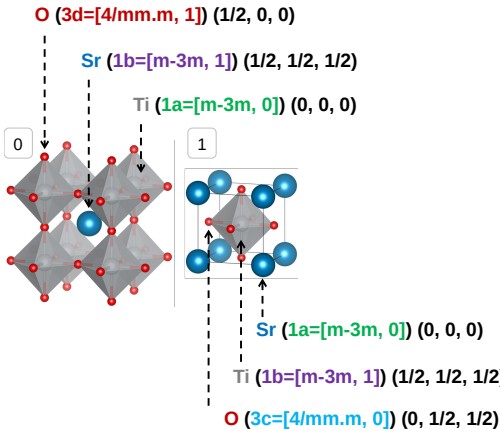

(a) A toy 2D crystal Goodall et al. (2020). It contains 4 mirror lines, and one rotation center. There are four Wyckoff positions, illustrated by shading. Magenta is the Wyckoff position that is invariant under all the transformations, it only contains a single point; red and yellow lie on the mirror lines, and teal is only invariant under identity transformation and occupies the rest of the space. Markers of the corresponding colors show one of the possible locations of an atom belonging to the corresponding Wyckoff position.

(b) Two possible equivalent Wyckoff representations of SrTiO₃, depending on the lattice center choice:
`[Ti, (m-3m, 0)], [Sr, (m-3m, 1)], [O, (4/mm.m, 1)]`
`[Ti, (m-3m, 1)], [Sr, (m-3m, 0)], [O, (4/mm.m, 0)]`

Figure 2: Wyckoff positions illustrations

2. Encoding Wyckoff positions using their universally–defined symmetry point groups and symmetry operations descriptors based on spherical harmonics; section 2.1.

3. Wyckoff Transformer architecture and training protocol that combine autoregressive probability factorization with permutation invariance; section 2.3.

4. Model invariance with the respect to the arbitrary choice of the coset representative of the space group affine normalizer; sections 2.1, 2.3.

5. Empirically, our model outperforms baseline methods in generating novel diverse materials conditioned on space group symmetry; section 3.2.

6. Despite not using the information about atom coordinates, our model achieves property prediction performance competitive with the machine learning models that use the full structure; section 3.2.

## 1.3 RELATED WORK

**Crystal generation** is a burgeoning field, with most state–of–the–art models using a differentiable non–invertible SO(3) invariant representation constructed from atom coordinates, such as a graph neural networks. Then they use diffusion or flow matching to solve the generation problem (Jiao et al., 2024a;b; Cao et al., 2024; Yang et al., 2023; Zeni et al., 2024; Xie et al., 2021; Klipfel et al., 2023; Luo et al., 2024; Sinha et al., 2024). Our approach uses discrete Wyckoff space, and fast autoregressive sampling, as compared to gradual refinement in the aforementioned works. WyFormer complements them naturally by providing symmetry constraints and/or initial structure approximation – the synergy with the most suitable partner, DiffCSP++, we evaluate thoroughly.

**Wyckoff positions and machine learning.** The concept of Wyckoff positions was originally published more than a 100 years ago (Wyckoff, 1922), which laid the groundwork for understanding equivalent positions in space groups, serving as a precursor to the International Tables for Crystallography. Given their elegant representation, naturally, in modern times WPs have found their way into machine learning. The main limiting factor in their adoption was the ability of machine learning algorithms to handle discrete structured data which is formed by WPs. WP–based representation was used for property prediction (Goodall et al., 2020; Jain & Bligaard, 2018; Möller et al., 2018; Goodall et al., 2022), and recently for generative models. Our work is inspired by Zhu et al. (2024), the first such model. It uses a VAE over one–hot–encoded information about WPs, as opposed our

Transformer encoder, a generally superior architecture for categorical data. AI4Science et al. (2023) use GFlowNet Bengio et al. (2023) to sample space group and chemical composition, but not the full Wyckoff representation. Finally, a concurrent work by Cao et al. (2024) independently explores a Transformer–based approach similar to ours.

The main difference between our and all other approaches, that are based on Wyckoff positions, is that they use Wyckoff letters as the representation. Wyckoff letter definitions depends on the space group, unlike site symmetry, leading to data fragmentation. Zhu et al. (2024); Cao et al. (2024) also don't take into account dependency of the Wyckoff letters on the arbitrary choice of the coset representative of the space group Euclidean normalizer. Finally, Cao et al. (2024) use positional encoding to establish the relationship between the chemical elements and Wyckoff positions they occupy, while we combine them in one token.

## 2 WYCKOFF TRANSFORMER (WYFORMER)

### 2.1 TOKENIZATION

Our work is based on the inductive bias that for stable materials space group symmetry and Wyckoff sites almost completely define the structure – more than 98% of the materials in MP–20 Xie et al. (2021) and MPTS–52 Baird et al. (2024) datasets, which tother contain almost all experimentally stable structures from the Materials Project Jain et al. (2013), have unique Wyckoff representations. Therefore is it safe to assume that for almost any Wyckoff representation there is either none, or just one stable material conforming to it. Symmetry captured by this discrete part is sufficient to determine properties of a material, such as piezoelectricity via non–centrosymmetry; direct/indirect band gap via positions of the valence/conduction bands in the Brillouin Zone, while the fractional coordinates can be linked to the magnitude of that property. We additionally prove this assumption by various predicting material properties, see section 3.2. Given a Wyckoff representation that reflects the lattice symmetry, coordinates can be determined as discussed in section 2.4.

We represent each structure as a set of tokens, as shown in figure 3. The first token contains the space group, the others chemical elements and WPs. We encode a WP as a tuple containing site symmetry and so-called *enumeration*. Several WPs can have the same site symmetry. To differentiate those WPs we enumerate them separately within each space group and site symmetry according to the conventional WP order Aroyo et al. (2006). For example, in space group 225 present in figure 3 WP 4a is encoded as (m-3m, 0), 4b as (m-3m, 1), and 8c as (-43m, 0). The purpose of this encoding is to take advantage of the fact that, unlike Wyckoff letters, site symmetry definition is universal across different space groups. We also develop a physics–based description of *enumerations* using spherical harmonics, it is discussed in details in Appendix B.

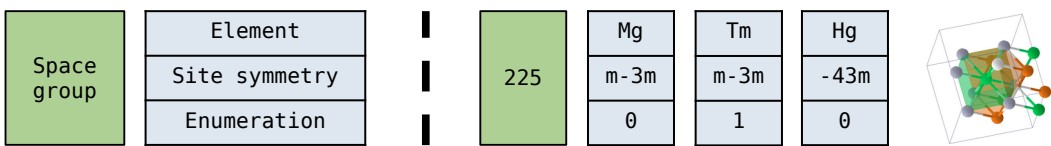

Figure 3: An example of structure tokenization, TmMgHg$_2$ mp-865981

The two-part encoding has another advantage. For some crystals *enumerations* part, and only this part, of Wyckoff representation is not uniquely defined, as it depends on the arbitrary choice of the coset representative of the space group Euclidean normalizer. See the example in figure 2b.

### 2.2 MODEL ARCHITECTURE

Elements, site symmetries, and enumeration are each embedded with a simple lookup table with trainable weights, the embeddings are concatenated; then we apply a linear layer. The reason is that in multi head attention different heads look at continuous blocks of the input vector.

Since our model is conditioned on space group, preventing data fragmentation is of utmost importance. To this end, space group is not encoded just as a categorical variable. Similarly to Bengio

et al. (2023) we use pyXtal to get one–hot–encoded $15 \times 10$ matrix that represents symmetry elements on each axis for each space group, flatten it, discard the positions that do not vary across the dataset and use the resulting vector as the space group embedding. Then we apply a linear layer, so the representation becomes learnable – but still transferable between space groups.

Token sequences are used as an input for a Transformer encoder Vaswani (2017); Devlin (2018). Wyckoff representation is permutation–invariant, so is Transformer; we don't use positional encoding, making the model formally permutation–invariant with the respect to the input.

**De novo generation** We use *enumerations* representation. We additionally add a `STOP` token to each structure. To represent states where some parts of token are known and others are not, we replace those values with `MASK`. We also add a fully–connected neural network for each part of the token that we want to predict, three in total. To get the prediction, we take the output of Transformer encoder on the token containing `MASK` value(s), concatenate it with a one–hot vector encoding presence in the input sequence of each possible value for this token part, and use it as the input for the corresponding fully–connected network.

**Property prediction** We use spherical harmonics representation. We take the average of the Transformer encoder outputs tokens, excluding the token corresponding to the space group, compute a weighted average with weights being equal to the multiplicities of WPs, and use the result as input for a fully–connected neural network that outputs a scalar predicted value.

## 2.3 TRAINING

Following approach by Wang et al. (2023); Abramson et al. (2024), we use a simple architecture and do no strictly enforce invariance with the respect to the choice of the coset representative of the space group affine normalizer, but rather leave it as a training goal by picking a randomly selected equivalent representation at every training epoch. It is especially viable because of the low number of variants; in MP–20 (Xie et al., 2021) dataset for 96% structures there are less than 10.

**De novo generation** We train the model to predict next part of a token in a cascade fashion: first the chemical element, then, conditioned on it, site symmetry and, finally, *enumeration*. On each training iteration we randomly sample known sequence length and the part of the cascade to predict; place `MASK` tokens as necessary, input the known parts of the sequences into the model, compute cross–entropy loss between the predicted scores and the target.

Unlike Transformer itself, auto–regressive generation is not permutation–invariant. The number of WPs is small, the average in MP-20 is just 3.0; this allows us to again follow the philosophy of Wang et al. (2023); Abramson et al. (2024) and train the model to be invariant with augmentation by shuffling the order of every Wyckoff representation at every training epoch. Moreover, we use multi–class loss when training to predict the fist cascade part, chemical element, further reducing learning complexity.

On MP–20 model is trained for $9 \times 10^5$ epochs using SGD optimizer without batching; due to the efficiency of the representation gradient backpropagation for the entire dataset fits into GPU memory. We use the loss on the validation dataset for early stopping, learning rate scheduling, and manual hyperparameter tuning.

**Property prediction** The model is trained using MSE loss with batch size 500, and Adam optimizer. For both MP–20 and AFLOW training takes around 5k epochs.

## 2.4 STRUCTURE GENERATION

We generate crystals conditioned on space group number which is sampled from the combination of training and validation datasets. Wyckoff representation is then autoregressively sampled using the Wyckoff Transformer. We use two ways to generate the final crystal structure conditioned on the representation, the details are described in appendix section C. They both start with sampling a structure conditioned on the Wyckoff representation with pyXtal (Fredericks et al., 2021), and then relaxing it with CrySPR (Nong et al., 2024) and CHGNet (Deng et al., 2023) or DiffCSP++ (Jiao et al., 2024b).

## 3 EXPERIMENTAL EVALUATION

### 3.1 DE NOVO GENERATION

#### 3.1.1 DATASET

We use MP-20 Xie et al. (2021), which contains almost all experimentally stable materials in Materials Project Jain et al. (2013) with a maximum of 20 atoms per unit cell, within 0.08 eV/atom of the convex hull, and formation energy smaller than 2 eV/atom, 45229 structures in total, split 60/20/20 into train, validation and test parts.

#### 3.1.2 METRICS

**Structure property similarity metrics**    Coverage and Property EMD (Wasserstein) distance, have been proposed as a low–cost proxy metric for de novo structure generation by Xie et al. (2021) and then followed by most of the subsequent work.

**Validity**    Xie et al. (2021) proposed verifying crystal feasibility according to two criteria:

- **Structural** validity means that no two atoms are closer than 0.5Å. All structures in MP–20 and almost all structures produced by state–of–the–art model fulfill it.
- **Compositional** validity means having neutral charge (Davies et al., 2019). Only 90% of MP–20 structures pass this test meaning that nonconforming structures are physically possible if somewhat rare.

**Novelty and uniqueness**    The purpose of de novo generation is to obtain new materials. Generating materials that already exist in the training dataset increases the model performance according to structure stability and similarity metrics, but such structures are useless for material design and just increase the gap between the proxy metrics and the model fitness for its purpose. Therefore we exclude generated materials that are not novel and unique from metric computation. On a deeper level, generative models for materials are subject to exploration/exploitation trade–off: the more physically similar are the sampled materials to the training dataset, the more likely they are stable and distributed similar to the data, but the less useful they are for the purpose of material design. From a purely machine learning point of view, novelty percentage serves a proxy metric for overfitting.

**Stability** is important as it determines whether the material, in fact, exists under normal conditions. It is estimated by computing energy above convex hull, and comparing it to a threshold $E_{\text{hull}} < 0.08$ eV, same as used during construction of MP–20 dataset. Then we compute S.U.N. Zeni et al. (2024) – the fraction of stable unique novel structures.

Due to DFT computational costs, we use CHGNet (Deng et al., 2023) for stability estimation of the generated structures, and then compute DFT for a manageable sample from the novel structures generated by the strongest models. Materials Project (Jain et al., 2013) is the source of the structures for the hull; we computed CHGNet predicted energies for it to use as references.

**Symmetry** of the structures has paramount physical importance. Controlling symmetries also leads to control over physical, electronic, and mechanical behavior, which is desirable in property–directed inverse design of materials. For example, in electronic materials, higher symmetry can improve carrier mobility and uniformity in electronic band structure, enhancing performance in applications such as semiconductors or optoelectronics. Furthermore, high–symmetry structures often exhibit isotropic properties, meaning their behaviors are the same in all directions, making them more versatile for industrial use. From a computational perspective, for a fixed set of atoms that constitute a crystal, enforcing symmetries (beyond the basic P1 translation symmetry) allows for computing permutations to search for useful materials while maintaining a focus on practical, synthesizable crystal structures.

This combination of stability, desirable properties, and computational efficiency makes symmetry consideration in crystals especially valuable in generative models for materials discovery. While higher symmetry is more tractable to compute, experimental realization could require external energy inputs (higher temperatures and pressures: think diamond vs graphite); most databases computed with DFT today are only at 0K and hence do not include this degree of freedom. Keeping

this in mind, to evaluate the models according to their ability to reproduce symmetry properties we propose four new metrics:

**P1** is the percentage of the structures that have symmetry group P1. In MP–20 the corresponding number is just 1.7%, and yet more than a third of the structures generated by some state–of–the–art models lack symmetry beyond lattice translation. We argue that presence of symmetry is good proxy value for structure feasibility that is difficult to capture in standard DFT computations, and would require finite–temperature calculations and/or improved methodologies.

**Novel Unique Templates (#)** is the number of the novel unique element–agnostic Wyckoff representations (section 2.1) in the generated sample. Element-agnostic means that we remove the chemical element, while retaining the symmetry information. For example, for the $TmMgHg_2$ in figure 3, it will be as follows:`{[(X, (m-3m, 0)), (X, (m-3m, 1)), (X, (-43m, 0))]; [(X, (m-3m, 1)), (X, (m-3m, 0)), (X, (-43m, 0))]}`. The metric provides a lower limit on overfitting and physically meaningful sample novelty: if two materials have different symmetry templates, their physical properties will be different, while inverse is not always true. It serves as an addition to the strict structure novelty, which provides the upper bound. Finally, the ability of a model to generate new templates allows it generate more structures before starting to repeat itself, as we demonstrate in Appendix I.

**Space Group $\chi^2$** is the $\chi^2$ statistic of difference of the frequencies of space groups between the generated and test datasets.

**S.S.U.N.** is the percentage of the structures that are symmetric (space group not P1), stable, unique and novel.

### 3.1.3 METHODOLOGY

**Wyckoff Transformer** was trained using MP–20 dataset following the original train/test/validation split. We sampled $10^4$ Wyckoff representations, then obtained $10^3$ structures using pyXtal+CHGNet and DiffCSP++ approaches described in section 3.1.3.

**WyCryst (Zhu et al., 2024)** only supports a limited number of unique elements per structure, therefore we trained it on a subsection of MP–20 containing only binary and ternary compounds, 35575 in total. Evaluation of Wyckoff Transformer trained on the same dataset as WyCryst is present in Appendix J. As WyCryst also produces Wyckoff representations, and not structures, the same pyXtal+CHGNet procedure was used to obtain them.

**CrystalFormer (Cao et al., 2024)** code and weights published by the authors were used by us to produce the sample, conditioned on the space groups sampled from MP–20.

**DiffCSP (Jiao et al., 2024a), DiffCSP++ (Jiao et al., 2024b), and FlowMM (Miller et al., 2024)** samples were provided by the authors.

Every data sample contained 1000 structures and was relaxed using CHGNet. The generated samples were filtered for uniqueness, more than 99.5% of structures for every method passed the filtering, therefore its impact is minimal and not further discussed.

We computed for DFT for $\sim 90$ novel structures for WyFormer and the baselines leading according to CHGNet–based metrics; detailed description of the settings is available in Appendix G.

### 3.1.4 DE NOVO STRUCTURE GENERATION RESULTS

Evaluation results are present in tables 1,2, and 3; a sample of generated structures is illustrated in figure 4.

Wyckoff Transformer achieves the best template novelty, fraction of asymmetric structures and space group distribution reproduction. Wyckoff Transformer and DiffCSP have similar S.S.U.N. (T–test $p = 0.8$) and S.U.N. (T–test $p = 0.2$). Given the limited DFT sample size, and DiffCSP's superior S.U.N. computed with CHGNet, it is likely that on a larger DFT sample it will surpass WyFormer.

The correlation of CHGNet–determined stability with DFT–determined is $0.33 - 0.44$, meaning that CHGNet is a blunt, but still useful tool for stability estimation.

Proxy metrics are present in table 3. Every model wins in at least one category, with the second place usually being close. We therefore would like to point out to some of the largest differences. WyCryst and CrystalFormer have significantly lower novelty compared to the other models. While manageable per se, it also means that the models have been overfitted, and their structures are more similar to the training dataset. DiffCSP++ oversamples the structures with the large number of unique elements, WyFormer matches the distribution most closely, as depicted in figure 6.

Table 1: Evaluation of the stability of the generated structures, as estimated by DFT and CHGNet. $E_{\text{hull}} < 0.08$ stability threshold is used, the same as in the training dataset, MP–20. Due to limited resources, DFT was only computed for the baselines with the strongest CHGNet S.U.N. and S.S.U.N.; # refers to the number of DFT samples; $r$ is the Pearson correlation between structures' stability determined by DFT and CHGNet. **Bold** indicates the values within $p = 0.1$ statistical significance threshold from the best.

| Method | | DFT ↑ | | $r$ | CHGNet ↑ | |
|---|---|---|---|---|---|---|
| | # | S.U.N. (%) | S.S.U.N. (%) | | S.U.N. (%) | S.S.U.N. (%) |
| WyFormer | 96 | 7.5 | 7.5 | 0.33 | 39.2 | 38.2 |
| WyFormerDiffCSP++ | 95 | **14.1** | **14.1** | 0.44 | 36.7 | 36.0 |
| DiffCSP++ | 94 | 8.5 | **8.5** | 0.32 | 41.4 | **40.8** |
| CrystalFormer | – | – | – | – | 33.9 | 33.8 |
| WyCryst | – | – | – | – | 36.6 | 35.2 |
| DiffCSP | 82 | **20.8** | 13.1 | 0.36 | **57.4** | 40.6 |
| FlowMM | – | – | – | – | 49.2 | 29.9 |

Table 2: Evaluation of the methods according to the symmetry metrics. Sample size is 1000; the metrics are computed only using novel structurally valid examples; structures were relaxed with CHGNet.

| Method | Novel Unique Templates (#) ↑ | P1 (%) ref = 1.7 | Space Group $\chi^2$ ↓ |
|---|---|---|---|
| WyFormer | 180 | 3.24 | 0.223 |
| WyFormerDiffCSP++ | **186** | **1.46** | **0.212** |
| DiffCSP++ | *10* | 2.57 | 0.255 |
| CrystalFormer | 74 | 0.91 | 0.276 |
| WyCryst | 165 | 4.79 | 0.710 |
| DiffCSP | 76 | 36.57 | 7.989 |
| FlowMM | 51 | 44.27 | 12.423 |

## 3.2 MATERIAL PROPERTY PREDICTION

MP–20 dataset contains two properties: formation energy and band gap, which we predict using WyFormer. The results are shown in Table 4. WyFormer achieves competitive results with the models that use full structures.

We also utilize the AFLOW database Curtarolo et al. (2012), which contains 4,905 compounds spanning a diverse range of chemistries and crystal structures. We use four material properties: thermal conductivity, Debye temperature, bulk modulus, and shear modulus. The data is divided into training, validation, and test sets using a 60/20/20 split.

On AFLOW, WyFormer demonstrated superior performance in predicting thermal conductivity. For the remaining three properties, the model's performance is comparable to that of the baseline models.

From this we can conclude that the symmetries and composition of the crystal alone already carry a considerable amount of information about its properties. This is especially true for the band

Table 3: Evaluation of the methods according to validity and property distribution metrics. Structures were relaxed with CHGNet. Following the reasoning in section 3.1.2, we apply filtering by novelty and structural validity, and do not discard structures based on compositional validity. An evaluation following the protocol proposed by Xie et al. (2021) is available in Appendix H.

| Method | Novelty (%) ↑ | Validity (%) ↑ | | Coverage (%) ↑ | | Property EMD ↓ | | |
| --- | --- | --- | --- | --- | --- | --- | --- | --- |
| | | Struct. | Comp. | COV-R | COV-P | $\rho$ | $E$ | $N_{\text{elem}}$ |
| WyFormer | 90.00 | 99.56 | 80.44 | 98.67 | 96.72 | 0.74 | 0.053 | **0.097** |
| WyFormerDiffCSP++ | 89.50 | 99.66 | 80.34 | 99.22 | 96.79 | 0.67 | 0.050 | 0.098 |
| CrystalFormer | _76.92_ | 86.84 | 82.37 | **99.87** | 95.13 | 0.52 | 0.100 | 0.163 |
| DiffCSP++ | 89.69 | **100.00** | **85.04** | 99.33 | 95.80 | **0.15** | 0.036 | 0.504 |
| WyCryst | _52.62_ | 99.81 | 75.53 | 98.85 | 87.10 | 0.96 | 0.113 | 0.286 |
| DiffCSP | **90.06** | **100.00** | 80.94 | 99.55 | 96.21 | 0.82 | 0.052 | 0.294 |
| FlowMM | 89.44 | **100.00** | 81.93 | 99.67 | **99.64** | 0.49 | **0.036** | 0.131 |

gap, where Brillouin zones are defined by symmetry, and thermal conductivity, which is a non-equilibrium phonon transport property also conditioned on underlying symmetry of the structure. To first order approximation kinetic theory, higher symmetry crystals typically have higher thermal conductivity due to (1) higher group velocities and (2) longer scattering times due to lower anharmonicity Newnham (2004); Yang et al. (2021).

Table 4: One–shot energy and band gap prediction. We computed CHGNet energy predictions on the MP-20 dataset, the rest of the baseline values are from Lin et al. (2023); The MP–20 test set is a part of CHGNet training set. Xie & Grossman (2018); Jha et al. (2019) report the error between DFT–computed and experimental results $\approx 0.08$ eV for energy, and $\approx 0.6$ eV for band gap.

| Method | Energy, meV | Band gap, meV | Train | Test |
| --- | --- | --- | --- | --- |
| CGCNN | 31 | 292 | | |
| SchNet | 33 | 345 | | |
| MEGNet | 30 | 307 | | |
| GATGNN | 33 | 280 | Materials Project–2018.6.1 | |
| ALIGNN | 22 | 218 | | |
| Matformer | 21 | 211 | | |
| PotNet | **19** | **204** | | |
| CHGNet | 34 | – | MPTrj | MP–20 |
| WyFormer | 25 | 247 | | MP–20 |

Table 5: MAE values for AFLOW dataset; baseline values are from Wang et al. (2021).

| Method | Thermal conductivity | Debye temperature | Bulk modulus | Shear modulus |
| --- | --- | --- | --- | --- |
| Roost | 2.70 | 37.17 | 8.82 | 9.98 |
| CrabNet | 2.32 | **33.46** | **8.69** | **9.08** |
| HotCrab | 2.25 | 35.76 | 9.10 | 9.43 |
| ElemNet | 3.32 | 45.72 | 12.12 | 13.32 |
| RF | 2.66 | 36.48 | 11.91 | 10.09 |
| WyFormer | **2.20** | 36.36 | 9.63 | 10.14 |

## 4    CONCLUSIONS AND LIMITATIONS

$E_{\text{hull}}$ determined from formation energy ($E_f$) as a proxy for stability is commonly used, but is imperfect, as it doesn't take into account configurational and vibrational entropic contributions, and hull determination relies on already known structures. Using CHGNet for stability estimation adds yet

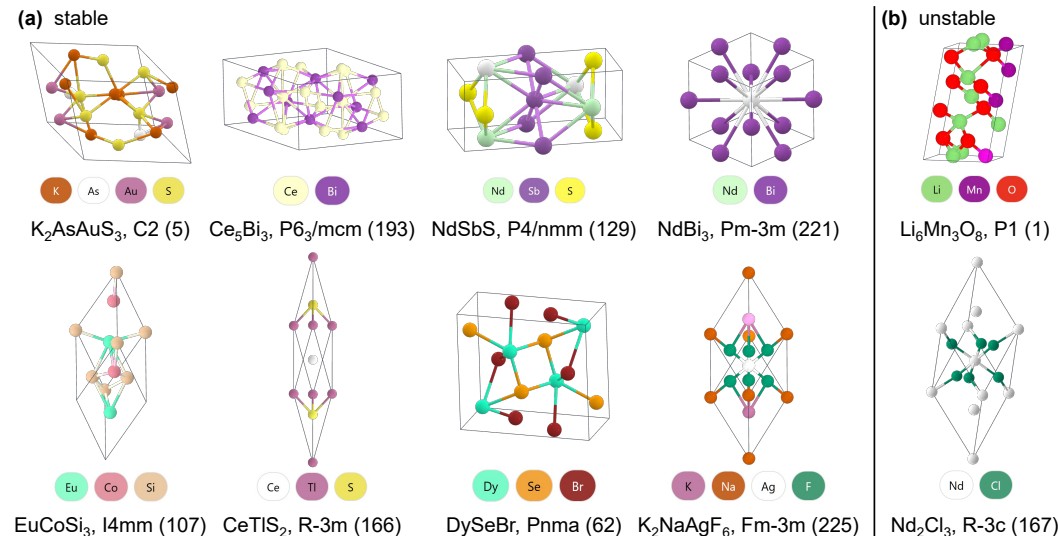

Figure 4: 10 structures generated from WyFormerDiffCSP++ and presented without additional relaxation. The labels contain the chemical formula, followed by the space group symbol in the short Hermann-Mauguin notation, and space group number. To the left 8 structures were randomly chosen from 15 stable structures as validated by DFT calculations, to the right 2 from unstable structures. The solid box lines represent the primitive cell.

another level of systematic uncertainty to these estimates. Moreover, our results, along with Miller et al. (2024) show that generated structures with space symmetry group P1 are consistently found stable at a much higher rate than they occur in nature. There are two logical conclusions from this: either DiffCSP and FlowMM have, in passing, discovered a new class of asymmetric materials – or our stability estimation methodology is systematically flawed.

Novelty and diversity evaluation is a crucial and, in our opinion, an open question. A model can generate structures that are same or similar to the ones in the training dataset, and are valid, but not very useful for material design. Counting complete duplicates is a step in the right direction, but doesn't measure substantial sample diversity Hicks et al. (2021).

An important part of the future work is Crystal Structure Prediction (CSP). Unlike the models that work with atoms and coordinates, it is hard to ensure that WyFormer output strictly conforms to a given stoichiometry. But we can add the stoichiometry as a generation condition, like space group. Then, as as we show in Appendix 6, WyFormer is four order of magnitude faster than other CSP solution, which allows to simply use rejection sampling.

In conclusion, we show that our Wyckoff Transformer represents a novel advancement in the generation of realistic symmetric crystal structures by leveraging Wyckoff positions to encode material symmetries more efficiently. Unlike previous methods, Wyckoff Transformer achieves a higher degree of structure diversity while maintaining stability, by encoding the discrete symmetries of space groups without relying on atomic coordinates. This unique tokenization of symmetry elements enables the model to explore a reduced, yet highly representative space of possible configurations, resulting in more stable and purportedly synthesizable crystals. The model respects the inherent symmetry of crystalline materials, outperforms existing models in generating both novel and physically meaningful structures. These innovations underscore the method's potential in accelerating material discovery while maintaining accuracy in predicting key properties like formation energy and band gap, comparable to complementary methods.

## REPRODUCIBILITY STATEMENT

The code and trained model weights will be published with the paper under an open source license.

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

APPENDIX

## A WYCKOFF REPRESENTATION WITH FRACTIONAL COORDINATES

A crystal can be represented as a space group, a set of WPs and chemical elements occupying them, the fractional coordinates of the WP degrees of freedom, and free lattice parameters. Such representation reduces the number of parameters by an order of magnitude without information loss. For example, see figure 5.

```
Group: I4/mmm (139)
Lattice: a = b = 8.9013,  c = 5.1991,  α = 90.0,  β = 90.0,  γ = 90.0
Wyckoff sites:
Nd @ [ 0.0000   0.0000   0.0000], WP [2a] Site [4/m2/m2/m]
Al @ [ 0.2788   0.5000   0.0000], WP [8j] Site [mm2.]
Al @ [ 0.6511   0.0000   0.0000], WP [8i] Site [mm2.]
Cu @ [ 0.2500   0.2500   0.2500], WP [8f] Site [..2/m]
```

Figure 5: Wyckoff representation of $Nd(Al_2Cu)_4$ (mp-974729), variable parameters in **bold**. If represented as a point cloud, the structure has $13[\text{atoms}] \times 3[\text{coordinates}] + 6[\text{lattice}] = 42$ parameters; if represented using WPs, it has just 4 continuous parameters (WPs 8i and 8j each have a free parameter, and the tetragonal lattice has two), and 5 discrete parameters (space group number, and WPs for each atom).

## B SPHERICAL HARMONICS

*Enumerations* are defined by an arbitrary convention, in this respect they are no better than Wyckoff letters. We propose a way to address this – a physics–based representation that is defined consistently across space groups. Consider a Wyckoff position consisting of a set of $k$ symmetry operations $\{A_i r + b_i, i = 1...k\}$. We apply those operations to points $r_1 = [0, 0, 0]$ and $r_2 = [1, 1, 1]$ obtaining two matrices $W^{(1)}$ and $W^{(2)}$: $W_i^{(j)} = A_i r_j + b_i r_j$. Finally, we convolve the transformed coordinates with spherical harmonics:

$$\phi_i^{(j)} = \arctan([W^{(j)}]_i^2, W^{(j)}]_i^1); \theta_i^{(j)} = \arccos([W^{(j)}]_i^3)$$

$$h^{(j)} = \sum_{i=1}^{k} |W_i^{(j)}|[Y_n^0(\theta_i^{(j)}, \phi_i^{(j)}), ..., Y_n^n(\theta_i^{(j)}, \phi_i^{(j)})]/k, \tag{1}$$

$n$ is the degree of spherical harmonics, a parameter, and the resulting complex vectors $h^{(1)}$ and $h^{(2)}$ each $n + 1$ dimensions. $n = 2$ is enough to disambiguate all Wyckoff positions with the same site symmetry belonging to the same space groups; $n = 1$ is not. Finally, we obtain the final $2n + 2$ dimensional descriptor $s$ by concatenation: $s = \Re(h^{(1)} \oplus h^{(1)}) \oplus \Im(\Re(h^{(1)} \oplus h^{(1)}))$ By itself harmonic representation does not allow for easy prediction, a way to use it for structure generation is discussed in Appendix N; performance is discussed in Appendix M.

## C STRUCTURE GENERATION DETAILS

The process of obtaining crystal structures from Wyckoff representations using PyXtal Fredericks et al. (2021) begins by specifying a space group and defining WPs. PyXtal allows users to input atomic species, stoichiometry, and symmetry preferences. Based on these parameters, PyXtal generates a random crystal structure that respects the symmetry requirements of the space group. Once the initial structure is generated, we then perform energy relaxation using CHGNet. CHGNet is a neural network–based model designed to predict atomic forces and energies, significantly speeding up calculations that would traditionally require density functional theory (DFT). We repeat the process for six random initializations and pick the structure with the lowest energy. Energy relaxation involves optimizing the atomic positions to reach a minimum energy configuration, which represents the most stable form of the material. CHGNet, trained on vast DFT datasets, can efficiently relax

crystal structures by adjusting atomic positions to reduce the total energy. This approach ensures that the final structure is not only symmetrical but also physically realistic in terms of energy stability.

For the 2nd structure generation method, DiffCSP++ is a diffusion-based crystal structure prediction model that focuses on generating purportedly stable crystal structures by sampling from an energy landscape in a physically consistent manner. DiffCSP++ generation also starts with PyXtal sampling.

## D  INFERENCE SPEED

We conducted experiments on a machine with NVIDIA RTX 6000 Ada and 24 physical CPU cores. For baselines, we used source code, model hyperparameters and weights published by the authors. Assuming that the downstream costs of structure relaxation by DFT or machine–learning interaction potential are fixed, the inference cost per S.U.N. structure is present in the table 6.

| Method | S.U.N. (%) | GPU ms per | | CPU s per | |
|---|---|---|---|---|---|
| | | structure | S.U.N. | structure | S.U.N. |
| WyFormerRaw | 4.8 | **0.05** | **1.0** | **0.105** | **2.2** |
| WyForDiffCSP++ | 14.1 | 840 | 5957 | 0.940 | 6.7 |
| DiffCSP | 20.8 | 360 | 1731 | 0.360 | 1.73 |
| DiffCSP++ | 8.5 | 1250 | 14705 | 1.35 | 15.9 |

Table 6: Inference time per S.U.N. structure. When a GPU is running, it also occupies a CPU core, which is taken into account. S.U.N. rates are measured according to DFT stability estimation. CHGNet is not used anywhere, for WyFormerRaw we sample a structure with pyXtal and use it directly as an input for DFT.

## E  PLOTS

Figure 6 contains the number of unique elements per structure for MP–20 and novel generated structures.

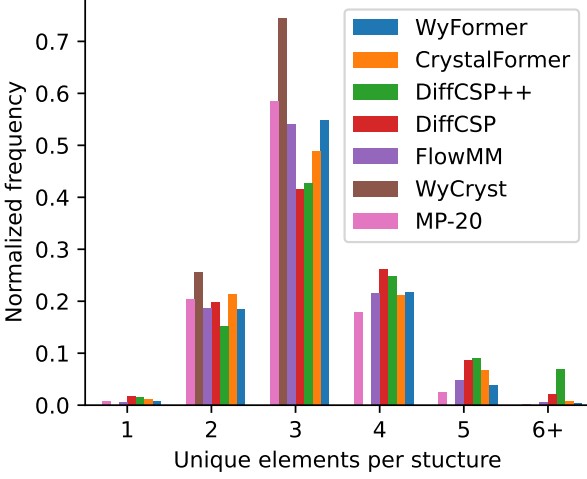

Figure 6: Distribution of the number of unique elements per structure for MP–20 and novel generated structures.

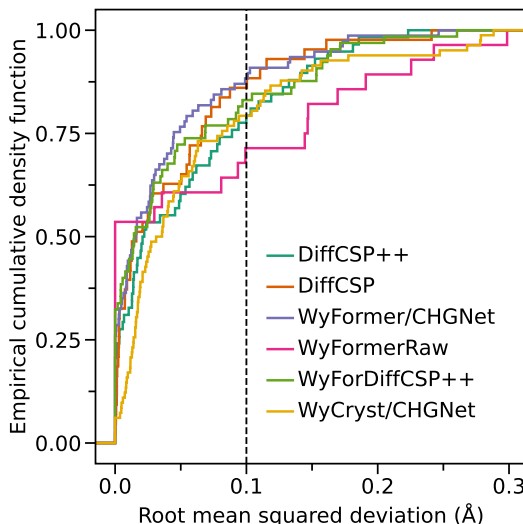

Figure 7: The empirical cumulative density function (ECDF) for root mean squared deviation (RMSD) of DFT-unrelaxed structures from DFT-relaxed counterparts. RMSD is calculated using `pymatgen.analysis.StructureMatcher` sub–module, in which only the RMSD of matched structure pairs is reported.

## F  ENERGY ABOVE HULL CALCULATIONS

To obtain the $E_{\text{hull}}$, we firstly constructed the reference convex hull data by querying all 153235 structures from the Materials Project (MP), and then using CHGNet (Deng et al., 2023) with using CrySPR interface (Nong et al., 2024) to do structure relaxations for all MP structures by relaxing both lattice cells and atomic positions (vc-relax), which renders 153,226 valid entries for relaxed structures and energies; secondly, for each 1,000 generated structures from each generative model, we followed the same vc-relax procedure to get the relaxed structures and energies; finally, using the `pymatgen.analysis.phase_diagram` sub–module the $E_{\text{hull}}$ for each entry of generated structure was computed by referencing to the MP convex hull, $E_{\text{hull}} = \max\{\Delta E_i\}$, where $\Delta E_i$ is the decomposition energy of any possible path for a structure decomposing into the reference convex hull.

## G  DFT DETAILS

All DFT structure relaxations were performed using the Vienna ab-initio simulation package (VASP) with the plane-wave basis set. Kresse & Furthmüller (1996) The electron-ion interaction is described by the projector augmented wave (PAW) pseudo-potentials. Kresse & Joubert (1999) The pseudo-potentials recommended by the VASP team are used. The exchange-correlation of valence electrons is treated with the Perdew-Burke-Ernzerhof (PBE) functional within the generalized gradient approximation (GGA). Perdew et al. (1996) The cutoff for kinetic energy of plane waves was set to 520 eV. Convergence thresholds of $10^{-8}$ eV for total energy and $10^{-4}$ eV $\text{Å}^{-1}$ atom$^{-1}$ for force were set. The Monkhorst-Pack scheme of $\boldsymbol{k}$-points sampling in the Brillouin zone with spacing of 0.15 $\text{Å}^{-1}$ is used Monkhorst & Pack (1976), in which the $\Gamma$ point is included. The Dudarev et al. simplified DFT+$U$ scheme  Dudarev et al. (1998) was adopted for the oxides and fluorides that contain one or more of the following transition metals: Co (3.32 eV), Cr (3.7 eV), Fe (5.3 eV), Mn (3.9 eV), Mo (4.38 eV), Ni (6.2 eV), V (3.25 eV), W (6.2 eV), consistent with the MP. Spin-polarized relaxations initialized with ferromagnetic, high-spin valence configurations were also performed to check if there is any magnetic atom with magnetism $\geq 0.15$ $\mu_{\text{B}}$.

The MP convex hull (v2023.11.1) was used as the reference hull. To do so comparably, additional DFT relaxations and self-consistent field (SCF) calculations using the VASP settings from `MPRelaxSet` and `MPStaticSet` in pymatgen were further performed based on the previously

relaxed structures. The raw total energies of SCF calculations using the `MPStaticSet` are then corrected using the correction scheme of `MaterialsProject2020Compatibility` before putting into the `PhaseDiagram` to obtain the DFT $E_{hull}$. What should be emphasized here is that the precision parameters, which are generated by `MPRelaxSet` and `MPStaticSet`, are too coarse, regarding especially the convergence thresholds ($2 \times 10^{-4}$ eV for energy, and $2 \times 10^{-3}$ eV $Å^{-1}$ for cumulative force) and the density of $\boldsymbol{k}$-points sampling (equivalent to a spacing of only 0.35 $Å^{-1}$). The `MPRelaxSet` is not strictly appropriate for direct structure relaxations for generated structures that typically are far off equilibrium.

## H LEGACY METRICS

For completeness sake, in table 7 we present the metrics computed following the protocol set up by Xie et al. (2021). We would like to again reiterate the issues with it. Firstly, the metrics are negatively correlated with structure novelty, the raison d'être for material generative models. Secondly, filtering by charge neutrality aka compositional validity means discarding viable structures. In terms of our newly defined metrics, let's consider stability and symmetry in detail below:

1. Stable is important as it determines thermodynamic stability of the generated structure and possibility of that compound to decompose to other energetically more favorable compounds,

2. Symmetry is critical to determine if the generated structure is not only stable, but also if there is a lower energy configuration belonging to a higher (or at least changed) symmetry. Therefore DFT relaxation for any generated structure is critical. If the symmetry changes after DFT relaxation, then the generated structure has a $E_{hull} < 0.08$ eV/atom (defined as Stable in #1 above) but there exists a higher symmetry structure which has lower energy - hence we need to pay attention to symmetry not changing after further DFT relaxation.

Table 7: Method comparison according the protocol set up by Xie et al. (2021).

(a) Directly using structures produced by the methods, without additional relaxation. Note that CHGNet is an integral part of generating structures with Wyckoff Transformer and WyCryst, so it's used.

| Method | Validity (%) ↑ | | Coverage (%) ↑ | | Property EMD ↓ | | |
|---|---|---|---|---|---|---|---|
| | Struct. | Comp. | COV-R | COV-P | $\rho$ | $E$ | $N_{elem}$ |
| WyckoffTransformer | 99.60 | 81.40 | 98.77 | 95.94 | 0.39 | 0.078 | 0.081 |
| WyFormerDiffCSP++ | 99.80 | 81.40 | 99.51 | 95.81 | 0.36 | 0.083 | **0.079** |
| CrystalFormer | 93.39 | 84.98 | 99.62 | 94.56 | 0.19 | 0.208 | 0.128 |
| DiffCSP++ | 99.94 | **85.13** | 99.67 | **99.54** | 0.31 | **0.069** | 0.399 |
| WyCryst | 99.90 | 82.09 | 99.63 | 96.16 | 0.44 | 0.330 | 0.322 |
| DiffCSP | **100.00** | 83.20 | **99.82** | 99.51 | 0.35 | 0.095 | 0.347 |
| FlowMM | 96.87 | 83.11 | 99.73 | 99.39 | **0.12** | 0.073 | 0.094 |

(b) All structures have been relaxed with CHGNet.

| Method | Validity (%) ↑ | | Coverage (%) ↑ | | Property EMD ↓ | | |
|---|---|---|---|---|---|---|---|
| | Struct. | Comp. | COV-R | COV-P | $\rho$ | $E$ | $N_{elem}$ |
| WyckoffTransformer | 99.60 | 81.40 | 98.77 | 95.94 | 0.39 | 0.078 | 0.081 |
| WyTransDiffCSP++ | 99.70 | 81.40 | 99.26 | 95.85 | 0.33 | 0.070 | **0.078** |
| CrystalFormer | 89.92 | 84.88 | **99.87** | 95.45 | 0.19 | 0.139 | 0.119 |
| DiffCSP++ | **100.00** | **85.80** | 99.42 | 95.48 | **0.13** | **0.036** | 0.453 |
| WyCryst | 99.90 | 82.09 | 99.63 | 96.16 | 0.44 | 0.330 | 0.322 |
| DiffCSP | **100.00** | 82.50 | 99.64 | 95.18 | 0.46 | 0.075 | 0.321 |
| FlowMM | **100.00** | 82.83 | 99.71 | **99.56** | 0.17 | 0.046 | 0.093 |

## I    TEMPLATE NOVELTY AND DIVERSITY

To asses the impact of template novelty on the diversity of the generated data can be assessed by evaluating the number of unique structures as the function of the total dataset size. We sampled 118k examples from the model with the lowest template novelty, DiffCSP++, and the highest, WyFormer. We present the number of unique samples as a function of the generated sample size in figure 8. DiffCSP++ uniqueness is clearly lower; due to its high inference costs (see Appendix 6), we were unable to prepare a larger sample.

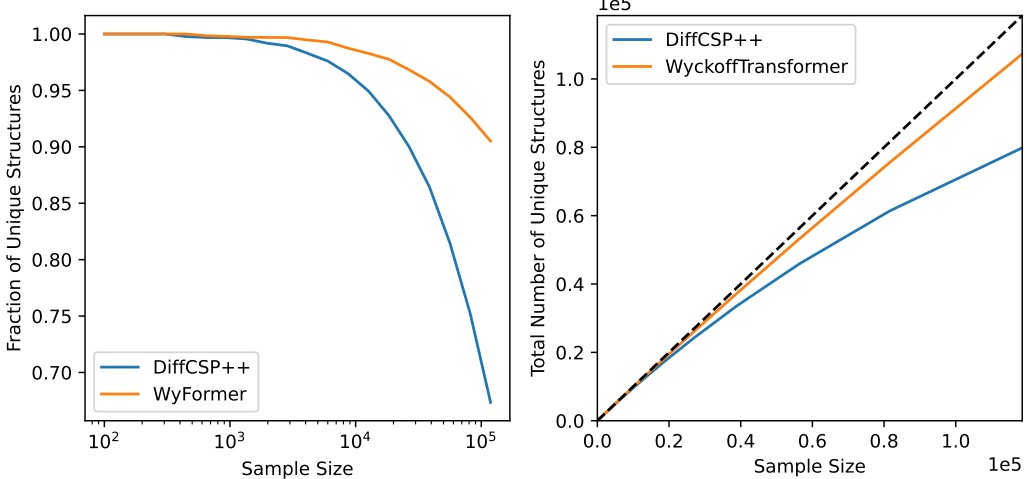

Figure 8: Fraction of unique structures and total number of unique structures as a function of sample size. For Wyckoff Transformer we used only the Wyckoff representations for uniqueness assessment, meaning that the uniqueness is likely to be slightly underestimated.

## J    EVALUATION ON MP−20 BINARY & TERNARY

Comparison of WyFormer to WyCryst is presented in tables 8 and 9. Both models were trained on a subset of MP−20 training data containing only binary and ternary structures, and similarly selected subset of MP−20 testing dataset is used as the reference for property distributions. All generated structures were relaxed with CHGNet. CHGNet was used for the formation energy computation for both generated and hull reference structures.

WyFormer outperforms WyCryst across the board. S.U.N. values are close, but this is achieved by WyCryst sacrificing sample diversity and property similarity metrics, with about half of the generated structures already existing in the training dataset.

| Method | Template Novelty (%) ↑ | P1 (%) ref = 1.7 | Space Group $\chi^2$ ↓ | S.S.U.N. (%) ↑ |
|--------|:---:|:---:|:---:|:---:|
| WyFormer | **25.63** | **1.43** | **0.224** | **37.9** |
| WyCryst | 18.51 | 4.79 | 0.815 | 35.2 |

Table 8: Evaluation of the methods according to the symmetry metrics. Aside from Template Novelty, metrics are computed only using novel structurally valid structures.

| Method | Novelty (%) ↑ | Validity (%) ↑ | | Coverage (%) ↑ | | Property EMD ↓ | | | S.U.N. (%) ↑ |
|--------|---------------|-------|-------|-------|-------|------|------|---------------|------|
| | | Struct. | Comp. | COV-R | COV-P | $\rho$ | $E$ | $N_{\text{elem}}$ | |
| WyFormer | **91.19** | **99.89** | **77.28** | **98.90** | **96.75** | **0.83** | **0.064** | 0.084 | **38.4** |
| WyCryst | 52.62 | 99.81 | 75.53 | 98.85 | 89.27 | 1.35 | 0.128 | **0.003** | 36.6 |

Table 9: Evaluation of the methods according to validity and property distribution metrics. Following the reasoning in section 3.1.2, we apply filtering by novelty and structural validity, and do not discard structures based on compositional validity. Validity is also computed only for novel structures.

## K  HYPERPARAMETERS

### K.1  OPTIMIZER

We use SGD optimizer with starting learning rate 0.2, and `ReduceLROnPlateau` scheduler with `factor=0.8` and patience of 40k epochs monitoring the validation dataset loss.

## L  FINE-TUNING LLM WITH WYCKOFF REPRESENTATION

To challenge Wyckoff Transformer's architecture, we compared it with pre–trained language models that were used in vanilla mode as well as after fine–tuning, essentially combining approach by Gruver et al. (2024) with Wyckoff representation. We explored two different textual representations of crystals corresponding to a given space group:

- **Naive**, which contains the specifications of atoms at particular symmetry groups encoded by Wyckoff symmetry labels: `Na at a, Na at a, Na at a, Mn at a, Co at a, Ni at a, O at a, O at a, O at a, O at a, O at a, O at a`

- **Augmented**, which contains the specifications of atom types with its' symmetries and site enumerations: `Na @ m @ 0, Na @ m @ 0, Na @ m @ 0, Mn @ m @ 0, Co @ m @ 0, Ni @ m @ 0, O @ m @ 0, O @ m @ 0, O @ m @ 0, O @ m @ 0, O @ m @ 0, O @ m @ 0`, where the set of valid symmetries is: `['2.22', '4/mmm', '1', '-3..', '6mm', 'm-3m', '2', '3mm', '.m', '-6mm2m', '4mm', '.32', '322', '.2/m.', '-1', '.m.', '..m', 'm.2m', '.3m', '3m', 'm2m.', '2mm', '-32/m.', '2..', '..2', '.3.', '2/m', '-43m', '4/mm.m', '.2.', '2/m2/m.', '23.', '222', 'm..', 'mm.', '-3.', 'm-3.', '3.', '4/m..', '.-3m', '2m.', '-32/m', '-42m', 'm.mm', '4..', 'm.m2', '422', '32.', '22.', '-622m2', '3m.', '.-3.', 'mmm..', '222.', 'mm2..', '-4m2', '2/m..', 'mm2', '-3m2/m', '-4m.2', '2mm.', '3..', '-42.m', '..2/m', '4m.m', '-4..', '6/mm2/m', 'm2m', 'm2.', '2.mm', 'mmm.', 'mmm', '32', 'm', '-6..']`

We fine-tuned the OpenAI `chatGPT-4o-mini-2024-07-18` model using different representations and compared it with the vanilla OpenAI `gpt-4o-2024-08-06` model. For each of the cases prompt looked like: `Provide example of a material for spacegroup number X`. The table below contains details of the model training:

Both training and inference times were measured using batch job execution on OpenAI's cloud. The fine-tuned model returned a JSON string that was easy to parse, while the vanilla model required additional parsing of its output.

Comparison the WyFormer to WyLLM is present in table 11. When fine–tuned, an LLM using Wyckoff representations shows similar performance to WyFormer – at a much greater computational cost. Using site symmetries instead of Wyckoff letters doesn't unequivocally increase the LLM performance, a possible explanation is that since this representation is our original proposition, the LLM is less able to take advantage of pre–training that contained letter–based Wyckoff

| Model | Base Model | Representation | Hyperparameters | Training Time | Inference Time | Number of Parameters |
|---|---|---|---|---|---|---|
| WyLLM-vanilla | gpt-4o-2024-08-06 | Naive | – | – | 74m | $\approx$ 200B |
| WyLLM-naive | gpt-4o-mini-2024-07-18 | Naive | epochs: 1, batch: 24, learning rate multiplier: 1.8 | 51m | 51m | $\approx$ 8B |
| WyLLM-site-symmetry | gpt-4o-mini-2024-07-18 | Site Symmetry | epochs: 1, batch: 24, learning rate multiplier: 1.8 | 95m | 37m | $\approx$ 8B |

Table 10: Comparison of different models and their characteristics. Number of parameters is not known exactly and is taken from public sources as an approximate estimation. For reference, WyFormer has 150k parameters.

| Method | Novelty (%) ↑ | Validity (%) ↑ | | Coverage (%) ↑ | | Property EMD ↓ | | |
|---|---|---|---|---|---|---|---|---|
| | | Struct. | Comp. | COV-R | COV-P | $\rho$ | $E$ | $N_{\text{elem}}$ |
| WyFormer | 89.50 | 99.66 | 80.34 | 99.22 | **96.79** | 0.67 | **0.050** | 0.098 |
| WyLLM-naive | 94.67 | 99.79 | 82.89 | 98.72 | 94.97 | 0.39 | 0.067 | **0.015** |
| WyLLM-vanilla | **95.59** | 99.82 | **88.75** | 94.46 | 59.67 | 2.23 | 0.234 | 0.253 |
| WyLLM-site-symmetry | 89.58 | **99.89** | 83.89 | **99.44** | 96.32 | **0.29** | nan | 0.039 |

| Method | Wyckoff Validity (%) ↑ | Novel Unique Templates (#) ↑ | $P1$ (%) ref = 1.7 | Space Group $\chi^2$ ↓ |
|---|---|---|---|---|
| WyFormer | **97.8** | 186 | 1.46 | 0.212 |
| WyLLM-naive | 94.9 | **237** | **1.38** | 0.167 |
| WyLLM-vanilla | 28.7 | 87 | 2.03 | 0.621 |
| WyLLM-site-symmetry | 89.6 | 191 | 2.24 | **0.158** |

Table 11: Comparison for WyFormer to different variant of WyLLM. All structures have been relaxed with DiffCSP++. Sample size is 1000 structures per model. The metrics described in section 3.1.2. nan is placed where the generated structures contained a rare element that crashed the property computation code. Wyckoff Validity refers to the percentage of the generated outputs that are valid Wyckoff representations. Aside from LLM–specific problems, such as non–existent elements, a Wyckoff representation can be invalid if it places several atoms at Wyckoff position without degrees of freedom, or refers to Wyckoff positions that do not exist in the space group.

representations. Without fine-tuning, the majority of LLM outputs are formally invalid, and the distribution of the valid ones doesn't match MP–20.

## M PERFORMANCE ANALYSIS OF ENCODING WPs WITH SPHERICAL HARMONICS

To assess impact of spherical harmonics we compare the performance of models with the same set of hyperparameters for the property prediction task on MP–20, leaving generative performance comparison for the future work. The results are present in table 12, hyperparameters in table 13.

| Representation | Energy MAE, meV | Band Gap MAE, meV |
|---|---|---|
| Site symmetry only | 31.7 | 247.8 |
| Wyckoff letter | 30.5 | 234.0 |
| Site symmetry & *Enumeration* | 30.7 | 244.1 |
| Site symmetry & Harmonics | 29.7 | 238.7 |

Table 12: Performance of WyFormer with different representation. The values are slightly different from table 4, as there we have tuned hyperparameters.

| Parameter | Value |
|---|---|
| Element embedding size | 16 |
| Wyckoff letter embedding size | 27 |
| Site symmetry embedding size | 16 |
| Site *enumerations* embedding size | 7 |
| Harmonic vector length | 12 |
| Batch size | 500 |
| Number of fully-connected layers | 3 |
| Number of attention heads | 4 |
| Dimension of feed–forward layers inside Encoder | 128 |
| Dropout inside Encoder | 0.2 |
| Number of Encoder layers | 3 |

Table 13: Hyperparameters used in the ablation study.

## N SAMPLING HARMONIC–ENCODED WPs

WP harmonic representation is a real–valued vector. But for each space group it can only take up to 8 possible values, so learning the full distribution of such vectors is not necessary. Therefore, we propose the following procedure:

1. Take the harmonic representations of all the WPs in all space group

2. Use K–means clustering to find 8 cluster centers.

3. Separately for each space group, assign harmonic labels to each *enumeration*:

   (a) Compute the Euclidean distances between all cluster centers and all WPs in the SG
   (b) Choose the smallest distance. Assign the WP to the corresponding cluster, remove WP and the cluster center from consideration.
   (c) Repeat until all WPs are assigned

This way all we obtain a discrete prediction target with one–to–one mapping with *enumerations*, but where physically–similar values are grouped together.

## O   SUPERCONDUCTOR CRITICAL TEMPERATURE PREDICTION

We used WyFormer to predict the critical temperature in superconductors on the 3DSC dataset Sommer et al. (2023); obtained test MLSE of 0.81

