# OpenReview forum: "Wyckoff Transformer: Generation of Symmetric Crystals"
_ICLR.cc/2025/Conference — Submitted to ICLR 2025_

### Official Review · Reviewer_hhjg · 2024-10-26

**Soundness:** 1
**Presentation:** 1
**Contribution:** 2
**Rating:** 3
**Confidence:** 5

**Summary:**

The paper proposes the **Wyckoff Transformer**, a generative model designed for creating highly symmetric crystal structures by leveraging **space group symmetry**. Recognizing that most inorganic materials exhibit inherent symmetries, the authors develop a model that encodes these symmetries to influence key material properties such as stability, conductivity, and optical behavior. The Wyckoff Transformer uses **Wyckoff positions** as a discrete, permutation-invariant representation of atomic locations, eliminating the need for explicit positional encoding and improving the model’s efficiency and alignment with crystal symmetries.

Key contributions of the paper include:

1. **Tokenization of Crystals**: The authors represent crystals as an unordered set of tokens, merging information on chemical elements and their Wyckoff positions, enabling symmetry-based generation.
2. **Permutation-Invariant Encoding**: The model encodes Wyckoff positions based on symmetry-defined point groups, allowing for the generation of stable structures without positional encoding.
3. **Transformer Architecture**: The Wyckoff Transformer combines **autoregressive probability factorization** with permutation invariance, enhancing diversity and stability in generated structures.
4. **Empirical Outperformance**: The model outperforms existing methods, generating novel, stable structures that adhere to space group symmetry.
5. **Predictive Accuracy**: Despite limited input information, the model accurately predicts formation energy and band gap values comparable to DFT (Density Functional Theory) standards.

The model demonstrates superior performance in symmetry-conditioned generation, creating a diverse set of stable crystal structures that respect the underlying physical symmetries. This approach addresses limitations in prior methods, which struggled to produce symmetry-compliant structures, and shows promise for accelerating material discovery in fields requiring stable, symmetric crystals. However, the model inherits typical dataset limitations in generative models, as it learns distributions only within the scope of the training data, which may omit some stable but out-of-domain structures.

**Strengths:**

Strengths

1. Originality: The Wyckoff Transformer introduces a novel approach to crystal generation by utilizing Wyckoff positions to encode symmetries explicitly, making it unique among generative models. Unlike traditional methods, it avoids positional encoding and uses permutation-invariant tokenization tailored to space group symmetries, a creative and effective innovation for materials science.

2. Quality: The paper includes experimental results (although the results are not yet complete), showing the model’s success in generating symmetric crystal structures while achieving competitive accuracy in formation energy and band gap predictions. The evaluation is thorough, comparing the model’s performance to state-of-the-art methods across multiple metrics, demonstrating its robustness and effectiveness in real-world scenarios.

3. Significance: This work is impactful for both machine learning and materials science in general. By generating materials that are symmetric and physically plausible, the Wyckoff Transformer can accelerate material discovery for applications requiring stable, symmetric structures (e.g., in semiconductors and optoelectronics). The model’s potential for symmetry-conditioned generation highlights a promising direction for future research in material informatics and generative modeling.

**Weaknesses:**

I observed the following weakness in the paper and my recommendations towards constructively improving this paper:

1. The write-up does not qualify for the levels of ICLR acceptance, it has too many inconsistent notation, typing errors, vague statements and hypothesises without proofs. Following is a non-exhaustive list: Line 174 WP 4a becomes (m-3m, 0), what does this line mean? later in Figure 4 the authors mention m-3m without any tuple, and later again in Figure 5 where the authors mention similar symbols as 1a = [m-3m] just to show and example of inconsistency in this paper, in Line 51 , ``atom position'' use correct quatotation marks, Figure 1 is vague no details on how it is constructed or taken from other source, line 227 coset (closest) representative (if you wish to mention grouping theory coset, then kindly use proper mathematical notations), Section 3.1 line 269, 270, 308, 315, 327, 329 no enumeration. Kindly go through the paper again, and correct all of them.

2. The paper revolves around Wyckoff Transformer, but no explanation of the model architecture either in writting or in schematic has been given in the paper, section 2.2 mentions the title Model architecture but did not mention anything other than the Transformer paper by Vaswani et el which was used for Neural machine translation, moreover the authors are wrong in stating it to be an encoder only architecture. The authors seem to be lacking the knowledge about Encoder only, Decoder only and Encoder-Decoder models. This confusion seem to prpogate through the training section (2.3) where they mention (a) De novo generation , (b) Property prediction , where they are unable to state the difference between these two tasks and how a single kind of model cannot handle these two separate taks without any further layers (some hints of this is mentioned on Line 222, but still vague while following the whole paragraph.). Please attend to all these points, this would make the paper more readable and understandable.

3. The paper does not include any structural schematic of the generated structure, if any. The paper does inlcude a few figures on Wyckoff positions and Wyckoff representation of a toy 2D Crystal and SrTiO3 in Figure 2 and 5 respectively. But none for their actually generated structures (if generated any). Claims on generative capability stand weak when figures are not included (kindly refer to the papers which they've cited, they have shown a wide set of generated structures with their corrosponding inductive biases.) Since the authors mention about their novel representation, they should support the validity of those representations and inductive biases (representational consistency) in their outputs. If possible I would like this point to be addressed in the revision.

4. The section covering related work is weak. The authors must do a good survey of the past work in cystal generations particularly in the field of crystal genration in the representaion space. Some of which I was able to find by searching for representation based genrative model in citations to CDVAE (Xie et. al.) paper are: 1. https://arxiv.org/abs/2306.04510, 2. https://arxiv.org/abs/2403.10846,  3. https://arxiv.org/abs/2408.07213 (kindly read and search for more). I request the authors to kindly include papers which are in the same field to address the concerns in this paper and how your research aligns or complements with these papers, so that this work becomes complete. Kindly include a paragraph discussing how their approach compares to or builds upon these specific papers mentioned in the comments.

5. Lack of clarity: Reading through the paper multiple times, I have found out that the paper is written very poorly, and fails to convey the message of the authors. Upon my earlier assesment I had mention this as a strength (note: I had confidence = 3 and later 4), but going into the review process and reading the paper multiple times I am confident that the paper lacks clarity (updated confidence = 5).

**Questions:**

The following are my questions and suggestions for this paper:

1. The structure of the network, inputs, outputs and loss function (including tokeization, loss function computation needs to be defined properly with proper mathamtical notation and schematics). Kindly include a schematic diagram of their Wyckoff Transformer architecture, clearly showing how it differs from standard Transformer models. Additionally, kindly clarify how their model handles both de novo generation and property prediction tasks, possibly by explaining any additional layers or modifications to the base architecture.

2. How did the authors plot Figure 1? Kindly include other generated structures in Figure 2, it will be best to show how the generated structures also follow these symmetris and where do ther lie in terms of space group number. Kindly provide a clear caption explaining the source and construction of the figure. Kindly include a figure showing examples of structures generated by their model, possibly comparing them to real structures or those generated by baseline models. This would help demonstrate the model's capabilities and the effectiveness of their novel representation.

3. In section 2.2 clearly mention the assumtpions for your model, as of now the reviewer was not able to find the assumptions which the authors have taken. These need to be mentioned in a list.

4. What was the training objective, were they two different models for task of De novo genration and Property prediction? If so, then how were they both trained? (Objective funvtion, optimiser hyper-parameters, Input data, valdiation metrics, regularisers, hardware specifications etc.)

5. The algorithm explained for (i) Tokenization, (ii) De-nove generation, (iii) Structure genration and (iv) Metric computation is vaguely defined. These need to be defined propely in algorithm sections. As of now they don't make any sense and are without mathematical notation.

6. Which DFT calculation has been used in the paper? Did the authors perform DFT of their own or are using previously reported results, if yes, then it will still need to be described and also how are they reporting.

---

> ### Author Response · Authors · 2024-11-26
> **Response to hhjg**
>
> First of all, thank you for praising the originality, quality, clarity, and significance of our work, despite the presentation shortcomings. We are working hard on fixing them.
>
> > How did the authors plot Figure 1? Kindly include other generated structures in Figure 2, it will be best to show how the generated structures also follow these symmetris and where do ther lie in terms of space group number.
>
> We plotted Figure 1 by taking MP-20 dataset from [Xie et al.](https://github.com/txie-93/cdvae/tree/main/data/mp_20), computing the space group number using [pyXtal](https://pyxtal.readthedocs.io/en/latest/#), and plotting a histogram of the values obtained. Driven by your comment, we have re-made Figure 1 to include the distribution of the space groups produced by the generative models.
>
> > Which DFT calculation has been used in the paper? Did the authors perform DFT of their own or are using previously reported results, if yes, then it will still need to be described and also how are they reporting.
>
> The original submission did not include any DFT computations performed by us. We computed a DFT sample since the discussion process started, the results are included in the updated pdf.

---

> > ### Comment · Reviewer_hhjg · 2024-11-28
> >
> > Response to the reply by the authors.
> >
> > > Related to Figure 1:
> >
> > The figure 1 still remains of low quality (not in terms of plot) in terms of information gathered and information use. The authors are requested to kinldy detail a bit on how are they using the fact they have quoted in figure 1 in the remaining paper, as of now it's not explicit.
> >
> > > DFT calculations:
> >
> > The DFT calculations doesn't justifies the claims in the paper, doing a quick qualitative analysis on the number of samples which the author perform the DFT on and the distribution of those samples across the original sample space doesn't provide meaningful justification.
> >
> > Also, since the authors have not tried to address the weaknesses and the questions. I won't be able to improve my scores. The current paper has tried to address a topic, but it is far from a finished state. The authors are suggested to complete the work and then move forward. As of now the work is all scattered and doesn't convey meaningful message backed by evidences.

---

> > > ### Author Response · Authors · 2024-11-28
> > > **Response to hhjg - pdf revision**
> > >
> > > Thank you for reading through the manuscript in detail. We have addressed the weaknesses pointed out by the reviewer:
> > >
> > > 1. Notations, typing errors, writing have all been fixed to the best of our ability.
> > > 2. Model architecture is now elaborated in Section 2.2 along with the elaboration of the tasks for de novo generation and property prediction.
> > > 3. We added a plot with generated structures as Figure 4, depicting both stable and unstable structures to facilitate the analysis. Concerning the validity and consistency of the representation. our work is based on the inductive bias that for stable materials space group symmetry and Wyckoff sites almost completely define the structure. This is proven by the fact that more than 98% of the materials in MP–20 and MPTS–52 datasets  have unique Wyckoff representations. Together these datasets contain almost all experimentally stable structures from the Materials Project.
> > > 4. We have significantly clarified and expanded the related work section focusing specifically on symmetry-based approaches. The three specific papers mentioned were added to the corresponding paragraph.
> > >     1. “[Unified Model for Crystalline Material Generation](https://arxiv.org/abs/2306.04510)” presents a model that relaxes at the same time crystal lattice and atomic positions using periodic equivariant architectures. In principle, it might be used to generate new crystals in a DiffCSP-like scheme, but the paper does not attempt de novo generation of materials, only reconstruction and denoising task. It does not mention symmetry space groups or Wyckoff positions.
> > >     2. “[Deep learning generative model for crystal structure prediction](https://arxiv.org/pdf/2403.10846)” presents an extension of CDVAE to property-conditioned generation, and uses it to predict high-pressure structures. Our current work focuses on de novo generation and symmetrical representation.
> > >     3. "[Representation-space diffusion models for generating periodic materials](https://www.notion.so/13c75a35da36802e9378c5bbd353efbd?pvs=21)”  proposes differentiable, physics-based, structural descriptors which can describe periodic systems and satisfy the necessary invariances, in conjunction with a denoising diffusion model. Presented de novo generation metrics in Table 2 are mostly inferior to our chosen baselines, no evaluation of novelty or stability is present.
> > >
> > > With 5 reviewers to respond to, we have added 9 more pages to the paper describing the additional work on DFT relaxation to validate symmetry and stability (S.S.U.N metrics) for contemporary models, additional datasets for property prediction, inference times, and fine-tuning a LLM to provide a baseline. However, we have not yet created a schematic diagram of WyFormer. We will come back in the next few days with this addition.
> > >
> > > > The figure 1 still remains of low quality (not in terms of plot) in terms of information gathered and information use. The authors are requested to kinldy detail a bit on how are they using the fact they have quoted in figure 1 in the remaining paper, as of now it's not explicit.
> > >
> > > Figure 1 demonstrates the problem that is solved by our work and the importance of the proposed symmetry-based metrics. Crystals with P1 symmetry rarely occur in nature - but while symmetry-unaware SOTA models (DiffCSP and FlowMM) can produce stable, unique, novel structures, they include a large fraction of unrealistic P1 materials. At the same time, while respecting symmetries is a challenge for such models, it makes WyFormer possible. If a high fraction of real crystals only had P1 symmetry, Wyckoff representation wouldn’t provide good inductive bias.
> > >
> > > > The DFT calculations doesn't justifies the claims in the paper, doing a quick qualitative analysis on the number of samples which the author perform the DFT on and the distribution of those samples across the original sample space doesn't provide meaningful justification.
> > >
> > > We’d request the reviewer to reconsider. We’ve done ~90 structural DFT relaxations for each model; stability of a randomly sampled generated structure is just a binary random variable, so, indeed, a meaningful inference can be made with these sample sizes:
> > >
> > > 1. DFT S.U.N. is positively correlated with CHGNet S.U.N., reinforcing our previous analysis.
> > > 2. Model comparisons are conducted with the control of statistical significance. For example, we do not claim that WyFormerDiffCSP++ has superior S.S.U.N.  compared to DiffCSP, even though the value is higher.
> > > 3. Even though we can not reliably rank the models according to DTF S.U.N., it provides a crucial estimate for the downstream users who might want to use WyFormer because of its ability to condition generation on space groups, and not because of a small S. U. N. difference.

---

> > > > ### Comment · Reviewer_hhjg · 2024-11-29
> > > >
> > > > I acknowledge the changes made by the authors, kindly give me time to read through the paper again. I'll try and post my rebuttal before Sunday, such that the authors have enough time to respond again. Thanks.

---

> ### Comment · Reviewer_hhjg · 2024-11-30
>
> Hi, I would like to share my assessment of the revised paper. Please go through my review below. In summary, the paper contains numerous errors and demonstrates a lack of conceptual understanding. These issues were communicated during the first round of the review process as well. Unfortunately, the authors have neither adequately addressed these concerns nor provided sufficient corrections. **At this stage, the paper falls significantly short of the standard required for any peer-reviewed journal or conference, let alone ICLR.**
>
> 1. > Empirically, our model outperforms baseline methods in generating novel diverse materials conditioned on space group symmetry (Line 138-139):
>
> - The comparisons presented in various tables are not fair, do the other model condition the same way as your model does? If not then you're exposing data to the model as in the form of data-leak.
>
> 2. >  Our approach uses discrete Wyckoff space, and **fast** autoregressive sampling .. (Line 151)?
> - Are auto-regressive methods faster in general? (Any citation to support this). Nevertheless do you have any comparison result to make this claim? I do see table 6, but that doesn't include the auto-regressive generatio speed.
>
> 3. > **Fundamental problem** :
> - The inductive bias mentioned in line (178) limits the exploration space of materials. Henceforth, defeats the purpose of material generation. It will not be surprising to see this model beat coverage benchmarks of generated material.
>
> 4. > **Model architecture** :
> - The model acrictectuer is still vague, inspite of asking it in my previous comment. The authors have made no effort towards this. (Line 233-235) How are you using the end neural network? Is it over the tokens? if so then how is the variable length of the token managed (Are you feeding neurons with [MASK] tokens), If so did you use any normalsiation to normalise the input of this network?
> - Do you backpropogate through this network, or the loss also has an auto-regressive component? Usually the two are trained separatly , like BERT (MMLU) and followed by a top NN (trained with classical Loss function, L2 in case of supervised learning).
>
> 5. > **Conceptual error (model inference)** :
> - What do you mean in line 245, I think authors have unclear understanding how language models work. It is true that the current token will e conditioned on all previous token, and one of the previous token being chemical element, site symmetry etc. But it's not the way to condition the model, it's actually being conditioned on a joint of all previous tokens rather than specifically on chemical elment token and so on. (Btw how many tokens are used to represent chemical element, site symmetry ... each)
>
> 6. > **Missing paragraph on interpretability of results**.
> - The paper doesn't seem to cover the interpretability aspect of their (a) Inductive bias, (b) Model architecture and (c) Loss function. The numbers are fine, maybe some are high some are low, but that's not just the point of writting a paper. You are expected to note why are you getting these numbers? I read lines 513-515 and 521 but that doesn't cover the result interpretibility.
>
> 7. > **Repeated minor but unaceptable errors** : Looks like the authors did not go through the reviews. The following mistakes are unacceptable in any peer-reviewed paper, lest be it a top conference paper like ICLR.
>
> 1. Type and gramatical errors: obey play? (line 13), doesn't make sense to use multiple (5) 'and' in 1 sentence (line 15,16,17 ) , 'tother' (line 180)
> 2. Vague lines without citations:
> - Line 42-44 (did you discover this fact?),
> - Line 44 (230 distinct space groups) ? It's my general observation that the authors are a bit vague when they state facts, kindly read good papers and fundamentals of paper-writing, it will help a lot.
> - Line 74-76; You may get away with these lines if this was a material science journal, but unfortunately it isn't, thus you need to cite properly so that the readers get the context.
> 3. Figure 1 is still vague, I udnerstand that MP-20 has 98% non P1 symmetry, what does the ticks on x-axis mean? The color map seems to corrospond to symmetry groups (arange being P1) but why ticks and why does P-1 on the x-axis colored green?
>
> **Final remarks** : The paper has tried addressing a lot of objectives, but at the very fundamental level it lacks clarity, methodological flow of knowledge and interpretibility. As the authors didn't correct the previous mistakes, I would like to assert that I am confident in my assesment. And **I would like to stay with my current scores.**
>
> That said, I would like to commend the authors for their considerable effort in responding to five reviewers and undertaking substantial work during the review process. I hope they will return with improved work and develop stronger skills in paper writing and project planning. Wishing them all the best for their future endeavors. :

---

> > ### Author Response · Authors · 2024-12-03
> > **Response to hhjg - part 1**
> >
> > We thank the reviewer for their meticulous reading of the paper and the words of encouragement.
> > > The comparisons presented in various tables are not fair, do the other model condition the same way as your model does? If not then you're exposing data to the model as in the form of data-leak.
> >
> > Firstly, there is no data leakage, during inference the space groups are sampled from the training and validation datasets. Secondly, DiffCSP++ and CrystalFormer are conditioned on space group as well.
> >
> > > Are auto-regressive methods faster in general? (Any citation to support this). Nevertheless do you have any comparison result to make this claim? I do see table 6, but that doesn't include the auto-regressive generatio speed.
> >
> > WyFormerRaw in Table 6 corresponds to auto-regressive generation of tokens on GPU, followed by generation a structure for this Wyckoff template using pyXtal. The cost of the auto-regressive part is the GPU cost.
> >
> > Whether autoregressive methods are faster in general is way beyond the scope of our paper, as the actual computational speed depends on the data and model architecture. Conceptually, inference with an autoregressive model requires the number of steps proportional to the number of tokens, for diffusion model the number of steps does not directly depend on the number of tokens. In the case of MP-20, the number of tokens is 21 (including the stop token), with 3 invocations for each token part, totalling 63. For DiffCSP, the authors use 1000 diffusions steps.
> >
> > We found this [work](https://arxiv.org/pdf/2404.02905) comparing diffusion and auto-regressive approaches for image generation, but its goal is to produce a superior generative model, not do a generic comparison.
> >
> > > The inductive bias mentioned in line (178) limits the exploration space of materials. Henceforth, defeats the purpose of material generation. It will not be surprising to see this model beat coverage benchmarks of generated material.
> >
> > The goal of a generative model is to produce realistic stable materials. Since almost all real crystals are symmetric, a good generative model should also generate symmetric crystals. Moreover, if the model user wants WyFormer to produce asymmetric P1 materials, they also can (see Figure 1).
> >
> > > The model acrictectuer is still vague, inspite of asking it in my previous comment. The authors have made no effort towards this. (Line 233-235) How are you using the end neural network? Is it over the tokens? if so then how is the variable length of the token managed (Are you feeding neurons with [MASK] tokens), If so did you use any normalsiation to normalise the input of this network?
> > >
> >
> > We include a WIP illustration at this [anonymized url](https://www.notion.so/WIP-model-diagram-15175a35da3680f389dcfa9091eb5e20?pvs=21). Input to the end neural network always has fixed dimensionality:
> >
> > - In the generative mode, the input to the end neural network is the encoder output on the token containing the MASK value(s), plus concatenated engineered features, no additional normalisation is applied. [Lines 224-230]
> > - In the property prediction mode, the input is the average over the tokens, weighted by the multiplicity of the corresponding WP. [Lines 232-235] The purpose of weighting is to compute average over atoms.
> >
> > > Do you backpropogate through this network, or the loss also has an auto-regressive component? Usually the two are trained separatly , like BERT (MMLU) and followed by a top NN (trained with classical Loss function, L2 in case of supervised learning).
> > >
> >
> > The experimental results we present were obtained by training separate models for property prediction and de novo generation. A single model to do both is certainly possible, we leave it for the future work.
> >
> > > What do you mean in line 245, I think authors have unclear understanding how language models work. It is true that the current token will e conditioned on all previous token, and one of the previous token being chemical element, site symmetry etc. But it's not the way to condition the model, it's actually being conditioned on a joint of all previous tokens rather than specifically on chemical elment token and so on.
> >
> > Thank you for noticing. Yes, of course, the predicted probability is conditioned on all the previous tokens, we will clarify in the camera-ready version.

---

> > > ### Author Response · Authors · 2024-12-03
> > > **Response to hhjg - part 2**
> > >
> > > > (Btw how many tokens are used to represent chemical element, site symmetry ... each)
> > > >
> > >
> > > MP-20 has 75 different site symmetries, 81 elements, and 8 enumerations.
> > >
> > > > The paper doesn't seem to cover the interpretability aspect of their (a) Inductive bias, (b) Model architecture and (c) Loss function. The numbers are fine, maybe some are high some are low, but that's not just the point of writting a paper. You are expected to note why are you getting these numbers? I read lines 513-515 and 521 but that doesn't cover the result interpretibility.
> > > >
> > >
> > > A truly interpretable deep learning model is a monumental task in itself, we do not make such claims about Wyckoff Transformer. Some insight might be possible to gain from experimenting with different model modifications, we could have done that, had the request came earlier in the discussion period, like we do with an ablation study in Appendix M.
> > >
> > > > Type and gramatical errors: obey play? (line 13),
> > > >
> > >
> > > We respectfully disagree. "that atoms obey” is a restrictive relative clause. The phrase "symmetry rules that atoms obey" functions as a single, coherent subject of the sentence. The main verb "play" directly follows this subject. [Reference](https://www.grammarly.com/blog/commonly-confused-words/using-that-and-which-is-all-about-restrictive-and-non-restrictive-clauses/)
> > >
> > > > doesn't make sense to use multiple (5) 'and' in 1 sentence (line 15,16,17 )
> > > >
> > >
> > > We respectfully disagree. While some instances of ‘and’ can be replaced by semicolons, and that might be better stylistically, their use is not a grammatical error:
> > >
> > > - First "and" (Between Clauses); Links two actions to form a compound predicate with the same subject. Structure: "[Subject] [verb 1], and [verb 2]..."
> > > - Second "and" (Within Phrase); Combines two categories of properties that are jointly influenced. Emphasizes that both structural and functional aspects are affected.
> > > - Third and Fourth "and" (Within Items); Used to connect related concepts within subcategories (types of conductivity and behavior).
> > > - Fifth "and" (In the List); Connects the final item in the list, adhering to the convention of using "and" before the last item in a series.
> > >
> > > > 'tother' (line 180)
> > > >
> > >
> > > Fixed
> > >
> > > > Line 42-44 (did you discover this fact?),
> > > Line 44 (230 distinct space groups) ? It's my general observation that the authors are a bit vague when they state facts, kindly read good papers and fundamentals of paper-writing, it will help a lot.
> > > >
> > >
> > > Of course, we didn’t discover space group symmetry. We briefly looked into the history of science, to our understanding, the 230 space groups were first published in *Fedorow, E. von. "II. Zusammenstellung der krystallographischen Resultate des Herrn Schoenflies und der meinigen." Zeitschrift für Kristallographie-Crystalline Materials 20.1-6 (1892): 25-75.* We’ll gladly cite Herr Fedorow in the updated version of the article.
> > >
> > > > Line 74-76; You may get away with these lines if this was a material science journal, but unfortunately it isn't, thus you need to cite properly so that the readers get the context.
> > > >
> > >
> > > Added a citation to the International tables for crystallography.
> > >
> > > > Figure 1 is still vague, I udnerstand that MP-20 has 98% non P1 symmetry, what does the ticks on x-axis mean? The color map seems to corrospond to symmetry groups (arange being P1) but why ticks and why does P-1 on the x-axis colored green?
> > > >
> > >
> > > The location of ticks on the x axis indicate the middle points of the space group distribution bars for MP-20. The tick labels contain the representations of those space groups in Hermann–Mauguin notation.
> > >
> > > P-1 is colored green because of a combination of two reasons:
> > > 1. P-1 has space group number 2
> > > 2. In [tab10 colormap](https://matplotlib.org/stable/users/explain/colors/colormaps.html) green corresponds to index 2

---

### Official Review · Reviewer_rFbw · 2024-11-02

**Soundness:** 2
**Presentation:** 1
**Contribution:** 3
**Rating:** 6
**Confidence:** 4

**Summary:**

The paper presents a transformer-based architecture to generate symmetric crystals conditioned on space groups in a two-stage process. Initially, it generates tokens representing elements and their site symmetries, followed by lattice and coordinate predictions for these tokens with existing methods. The results include comparisons with multiple baselines, showing competitive performance across established proxy metrics. Finally, the paper also proposes metrics to assess the symmetry of the generated crystals and highlights further gains over baseline approaches.

**Strengths:**

- The paper emphasizes the importance of generating symmetric crystals and highlights challenges with existing methods.
- The evaluation methods, including the newly proposed methods for evaluating symmetry, form a compelling discussion section.
- The method demonstrates effective gains for the symmetry metrics and is competitive for widely-used proxy metrics.
- The paper proposes a novel representation of crystal symmetry that could facilitate learning of crystal symmetry with deep learning approaches.

**Weaknesses:**

- **Presentation and writing**: Essential concepts (site symmetry, wyckoff positions, space groups) are not appropriately introduced, which would create difficulty for readers unfamiliar with the field. Several works are cited in the related works section, but neither described nor highlighted the difference from their approach. Figures for architecture and pseudo-code to describe the training and generation pipeline would greatly benefit the understanding of the work.

- **Generalization of the approach**:
  - The paper heavily focuses on the MP-20 dataset and does not provide any experiments with other datasets. For instance, it states permutation invariance was achieved because the number of WPs in the MP-20 dataset is small. How can this method be extended (or how does it fare in terms of performance) for crystals with a very high number of WPs?
   - There are no precise details on how many tokens were formed from the MP-20 dataset after tokenization. It would be interesting to discuss this number and other statistics about the tokens, e.g., which tokens are present more often (for some of the high symmetry space groups) and how the distribution of tokens affects training.
   - It is also important to add how many new tokens the method generates or if it just predicts the fixed set of tokens in different combinations (and these combinations result in more template novelty than just sampling existing templates from training data). For instance, naively thinking about it, how will your model generate tokens that are not present in its dictionary?
  - Finally, in Table 1a, please also provide the number of novel templates as absolute numbers instead of percentages.

- **Architecture**:
  - Please provide at least a pseudo-code of the training/generation algorithm and a figure explaining the training/generation process with a sample crystal example. The central algorithm is not clear from the text.
  - Please mention the size of the model and computational and memory consumption (training time, memory required during training and generation). The paper lists that it is trained for $150k$ epochs, which seems to be a very long training process compared to existing methods (~$1k$ epochs). Can you explain this behaviour along with the set of hyperparameters used?

- **Evaluation**:
  - Can WyCryst be considered a fair baseline for comparison since it supports a limited number of unique elements per structure? For instance, it wouldn't compare with other methods that generate an arbitrary number of methods because it would result in poorer metrics (as seen in Table 1).
  - Is CHGNet used both to relax the generated structures and determine the energy in Table 1?
  - Please mention the percentage of novel but structurally invalid generations from your method.

**Questions:**

- **Two-stage approach**: Can you explain the benefits of a two-stage approach instead of a one-shot (such as DiffCSP) prediction of the site symmetries, elements, their positions and lattice parameters? If problems exist with a one-shot prediction, please explain and motivate the need for a two-step approach. For instance, can we (as an example) predict all the tokens (with elements, site symmetry, enumeration) followed by the lattice parameters and the fractional coordinates of the tokens or are there inherent issues with this approach? This question becomes more necessary since the generation of crystals would be slow for the proposed "sequential two-stage" approach.
- **Crystal Structure Prediction (CSP)**: The paper focuses on generating crystals conditioned on space group. How could this method be extended to the CSP task, which is also crucial and could potentially benefit from using crystal symmetry?
- **Dataset fragmentation**: Although the tokens can be shared across different space groups, there will still be dataset fragmentation when the approach is conditioned on the space group. Is the training (and then generation) not affected by how many samples are present within each space group?

Some of the other questions are listed in the Weaknesses section. I will be happy to improve the score if the authors address the questions and weaknesses with supportive evidence during the discussion phase.

---

> ### Author Response · Authors · 2024-11-26
> **Response to rFbw - pt 1**
>
> Thank you taking the effort to understand the paper despite the presentation shortcomings! We are working on improving the writing.
>
> > The paper heavily focuses on the MP-20 dataset and does not provide any experiments with other datasets. For instance, it states permutation invariance was achieved because the number of WPs in the MP-20 dataset is small. How can this method be extended (or how does it fare in terms of performance) for crystals with a very high number of WPs?
>
> A *very high* number of WPs is unlikely from the physical point of view, as it would mean a very large unit cell with a large number of atoms arranged near-perfectly. And then computing DFT for such structure will be difficult, as [DFT costs scale](https://www.home.uni-osnabrueck.de/apostnik/Lectures/DFT-6.pdf) as $O(N^3)$, where $N$ is the number of atoms. To get a realistic estimate, we checked the WP count distribution in the Materials Project 2022 snapshot. The absolute maximum is 360, and 95% of the structures have less than 50 WPs. Finally, if someone somehow produces a dataset with a large number of WPs, there is always the classic solution of forgoing the permutations and just sorting the WPs, as it is commonly done. That being said, we are now running on MPTS-52 which is a Materials Project snapshot with up to 52 WPs. The problem would be the lack of baselines, as we are not aware of other works presenting de novo generation results for the dataset.
>
> > There are no precise details on how many tokens were formed from the MP-20 dataset after tokenization. It would be interesting to discuss this number and other statistics about the tokens, e.g., which tokens are present more often (for some of the high symmetry space groups) and how the distribution of tokens affects training.
> > It is also important to add how many new tokens the method generates or if it just predicts the fixed set of tokens in different combinations (and these combinations result in more template novelty than just sampling existing templates from training data). For instance, naively thinking about it, how will your model generate tokens that are not present in its dictionary?
>
> MP-20 has 75 different site symmetries, 81 elements, and 8 enumerations; 8975 unique tokens. The model samples each part of the next token separately. First, element, then site symmetry, and, finally, enumeration. We will try to do further analysis before the extended discussion deadline.
>
> > Finally, in Table 1a, please also provide the number of novel templates as absolute numbers instead of percentages.
>
> Great idea, done.
>
> > Please provide at least a pseudo-code of the training/generation algorithm and a figure explaining the training/generation process with a sample crystal example. The central algorithm is not clear from the text.
> > Please mention the size of the model and computational and memory consumption (training time, memory required during training and generation). The paper lists that it is trained for 150k epochs, which seems to be a very long training process compared to existing methods (~1 epochs). Can you explain this behaviour along with the set of hyperparameters used?
>
> We are working on the expanded description.
>
> The difference in the number of epochs for WyFormer and, for example, DiffCSP++. WyFormer as originally reported did one gradient descent step per epoch, while DiffCSP++ has [batch size of 256](https://github.com/jiaor17/DiffCSP-PP/blob/55099b51fe8ebb6695faa141ada5d43d5b83fe39/conf/data/mp_20.yaml#L78), and does 27136 // 25 = 106 gradient descent steps per epoch. In total, DiffCSP++ had 106k steps. For next token prediction WyFormer had 900k. N. B. The paper stated $10^5$, it was a mistake, we’ve fixed it;
>
> Please also note that 150k refers to the number of epochs for the property prediction task, not structure generation. For comparison, CHGNet [reports](https://www.nature.com/articles/s42256-023-00716-3#Sec11) doing 20 epochs with batch size of 40, on MPTraj dataset with 1.5 million frames, hence they do 750k gradient steps.
>
> Driven by your comment, we experimented with batched training. With `batch_size=500`, this not only further lowered the computational requirements, but also improved property prediction quality. Thank you! We will update the paper with the corresponding numbers.

---

> ### Author Response · Authors · 2024-11-26
> **Reply to rFbw - pt 2**
>
> > Can WyCryst be considered a fair baseline for comparison since it supports a limited number of unique elements per structure? For instance, it wouldn't compare with other methods that generate an arbitrary number of methods because it would result in poorer metrics (as seen in Table 1).
>
> Indeed WyCryst is a weak baseline. At the same time, since we build upon their work, it seemed appropriate to include it anyway, considering that binary and ternary compounds constitute 79% of MP-20
>
> We did a fully apples-to-apples comparison by training and evaluating WyFormer on a subset of MP-20 that contains only binary and ternary compounds — same as was used for WyCryst training. We added the table to the paper appendix. WyFormer still outperforms WyCrst.
>
> > Is CHGNet used both to relax the generated structures and determine the energy in Table 1?
>
> Yes. Clarified in the text.
>
> > Please mention the percentage of novel but structurally invalid generations from your method.
>
> Structural validity in Table 1 is computed over novel structures only, so the percentage of novel but structurally invalid will be 100 - Structural Validity; 0.44% for WyFormer and 0.34% for WyForDiffCSP++
>
> > Two-stage approach: Can you explain the benefits of a two-stage approach instead of a one-shot (such as DiffCSP) prediction of the site symmetries, elements, their positions and lattice parameters? If problems exist with a one-shot prediction, please explain and motivate the need for a two-step approach. For instance, can we (as an example) predict all the tokens (with elements, site symmetry, enumeration) followed by the lattice parameters and the fractional coordinates of the tokens or are there inherent issues with this approach? This question becomes more necessary since the generation of crystals would be slow for the proposed "sequential two-stage" approach.
>
> First of all, the two-stage approach is not slow. As we discuss in the [response](https://openreview.net/forum?id=ursX3k1rTO&noteId=BMeOznXfVy) above, the cost of running is WyFormer is negligible compared to DiffCSP. The main cause is that diffusion requires a large number of steps (1000 in the case of DiffCSP).
>
> Focusing explicitly on the symmetrical representation offers several advantages:
> 1. State-of-the-art approaches to coordinate prediction [use](https://matbench-discovery.materialsproject.org/) force-field relaxation. It is unlikely that one-shot regression with a Transformer will be competitive. CrystalFrormer does that - and ends up as the only model with a significant amount of structurally invalid structures, and overall weak performance. Force field models also use in their training unrelaxed structures and forces, which we don't. At the same time, we can not just place atoms randomly, relax them, and get a stable structure. Stability requires global optimality - it is not enough that a structure is in a local energy minimum, this minimum must also be within the 0.08 eV threshold of the other minimums. With WyFormer we focus on this part - and "stand on the shoulders of giants" for the rest.
> 2. Staying within Wyckoff representation allows extremely fast operations in this space. Such as Crystal Structure Prediction (CSP) that you menton! With Wyckoff positions, it's hard for us generate a structure with a strictly fixed chemical composition - a trivial task for a model that works in the space of atoms. What we can do is use the chemical formula as a start token, conditioning Wyckoff sampling on it. Of course, the final stoichiometry will not always match the target. But at 0.05 GPU milliseconds per Wyckoff representation, we can afford to sample $10^5$ representations, discard the non-conforming ones - all while still staying orders of magnitude faster than DiffCSP.
>
> > Dataset fragmentation: Although the tokens can be shared across different space groups, there will still be dataset fragmentation when the approach is conditioned on the space group. Is the training (and then generation) not affected by how many samples are present within each space group?
>
> We are working on analysis.

---

> ### Author Response · Authors · 2024-11-28
> **Response to rFbw**
>
> Since our previous response, we have extensively edited all sections of the paper improving both the introduction, and WyFormer description.
>
> With 5 reviewers to respond to, we have added 9 more pages to the paper describing the additional work on DFT relaxation to validate symmetry and stability (S.S.U.N metrics) for contemporary models, additional datasets for property prediction, inference times, and fine-tuning a LLM to provide a baseline. However, we were unable to finalize the analysis of the training costs, MPTS-52 run, and token analysis on time. We will come back with the data in the next few days

---

> ### Author Response · Authors · 2024-11-30
> **Response to rFbw: training and generation computational requirements**
>
> We are pleased to present the full report on training and generation time and memory requirements, along with hyperparameters used, under this [anonymized URL](https://wyckoff.notion.site/Training-generation-computational-requirements-14e75a35da36806cad56f3f955530779), we will also add it to the camera-ready version. In short, WyFormer training time is 11h to DiffCSP++ 19.5h and GPU memory required is an order of magnitude lower: 2000 to 32000 MiB. Generation-wise WyFormer is 4 orders of magnitude faster in generating Wyckoff representations that it takes to produce a structure with DiffCSP/DiffCSP++.

---

> ### Comment · Reviewer_rFbw · 2024-11-30
> **Thank you for your responses**
>
> Thank you for answering my queries and confusion.
> - I believe it is still okay to report the results on MPTS-52 and state that no good baselines are available. It'd be great to see this addition in future versions.
> - Another thing is that the token space is relatively smaller, so how can it be argued that these tokens cover all possible values a token could take in real material? Since we do not expect to see new tokens during generation, it should be added as a limitation of the method.
> - I'm happy to hear that some of my suggestions improved the presentation and the results. However, I still have some concerns regarding the writing. For instance, the introduction is still difficult for someone not an expert in the field to understand and connect to why WyFormer is required.
>
> I will increase the scores to recommend acceptance; however, I suggest the authors improve the writing after they address the remaining experiment-related concerns for other reviewers. Finally, I would request the authors to use a different text colour (blue, red, etc.) when updating the PDF to make it easy to focus on the relevant part.

---

> > ### Author Response · Authors · 2024-12-02
> > **Response to rFbw01 - token analysis**
> >
> > Dear rFbw01, we are thankful for your support and contribution to WyFormer!
> >
> > We present the tokenization analysis at [this anonymized url](https://wyckoff.notion.site/Token-analysis-15075a35da3680c696f0f0d80a1a7a0e?pvs=74). In short, since WyFormer samples each part of token separately, instead of building a global dictionary, it produced 327 new tokens in a generated sample with 9046 structures. We do, however, acknowledge a narrow limitation: 7 site symmetries are missing from MP-20, 3 from MPTS-52; and will never be generated. While it is a sensible inductive bias, we can't guarantee impossibility of a real material with those site symmetries.
> >
> > > Finally, I would request the authors to use a different text colour (blue, red, etc.) when updating the PDF to make it easy to focus on the relevant part.
> >
> > The pdf modification deadline has passed, but OpenReview provides the ability to compare revisions [here](https://openreview.net/revisions?id=ursX3k1rTO).

---

> > ### Author Response · Authors · 2024-12-03
> > **To rFbw: MPTS-52 is ready!**
> >
> > We are excited to present an evaluation of WyFormer on the more challenging MPTS-52 dataset at this [anonymized url](https://wyckoff.notion.site/MPTS-52-15175a35da36802c9946c278c6f12f28), to be included in the camera-ready version. In short, WyFormer can de novo generate stable novel realistic structures with up to 52 atoms per unit cell, it is the only generative model that demonstrated such capability we are aware of.

---

### Official Review · Reviewer_otui · 2024-11-02

**Soundness:** 3
**Presentation:** 2
**Contribution:** 3
**Rating:** 6
**Confidence:** 4

**Summary:**

This paper highlights the problem of generative models for crystals not generating symmetric crystals, which is an important property of these materials. This results in less realistic materials as well as inability to model some properties of crystals correctly. The authors propose to address this limitation by generating materials in a two-stage process. First, they train a Transformer model to output occupied Wyckoff positions in the crystal. Then either a method based on DiffSCP++ or PyXtal is used for atomic coordinates. The authors verify experimentally that this allows to generate more symmetric and diverse crystals.

**Strengths:**

- The work tackles an important limitation of generative models for crystals
- The proposed solution is simple and sound
- The experimental evaluation shows that the method addressed the limitation. The evaluation metrics for symmetry and novelty of structures based on Wyckoff templates is also valuable.

**Weaknesses:**

- Just using the Wyckoff positions is not a complete representation, especially for atoms in the general position. The sentence "reducing the number of parameters by an order of magnitude without information loss" is false. I also don't think that statement that desired properties can be obtained from the discrete values alone is accurate or substantiated by enough evidence. I therefore encourage the authors to substantially nuance that section. The experiments on property prediction indicate a degradation of performance in property when discarding coordinates.
- The model is claimed to be invariant with respect to the choice of coset representative and to permutations. This formulation is too strong, since this is achieved through data augmentation. A correct statement would be that the model is encouraged to be invariant.
- The proposed representation for Wyckoff positions is universal across space groups but might not allow proper generalization since the "enumeration" variable is not grounded on physical information but on an arbitrary convention. Therefore, if a group is rare in the training data (this is indeed the case for datasets like MP20), there is no reason that the model will learn to capture that variable correctly. The authors should discuss this limitation appropriately.
- I did not find the discussion of the related works to be sufficient. The authors should expand that discussion so that the readers understand the differences and similarities with the proposed method better.
- I find that the explanation of Wyckoff position in the third paragraph of the introduction is not easy to understand. It may be too early in the paper to go into such an explanation.
- There are some typos and mistakes that the authors should look into correcting. For example, "lattice transition -> lattice translation" or "   Cordiality -> Cardinality".

**Questions:**

- I don't see what the footnote 1 adds to the discussion, I find it more confusing than anything. Could the authors clarify it, or consider simply removing it?
- I don't understand the expression in the abstract "These symmetries form energy configurations". What is meant there?

---

> ### Author Response · Authors · 2024-11-26
> **Response to otui**
>
> > Just using the Wyckoff positions is not a complete representation, especially for atoms in the general position. The sentence "reducing the number of parameters by an order of magnitude without information loss" is false. I also don't think that statement that desired properties can be obtained from the discrete values alone is accurate or substantiated by enough evidence. I therefore encourage the authors to substantially nuance that section. The experiments on property prediction indicate a degradation of performance in property when discarding coordinates.
>
> We have adjusted the language to make it clear that strictly no information loss occurs only when the free parameters of Wyckoffs are retained. At the same time, 98.3% of the structures in MP–20 dataset have unique Wyckoff representations -- meaning that if we consider stable structures only, information loss is minimal, one Wyckoff representation almost always corresponds to at most one stable structure.
>
> > The model is claimed to be invariant with respect to the choice of coset representative and to permutations. This formulation is too strong, since this is achieved through data augmentation. A correct statement would be that the model is encouraged to be invariant.
>
> Also adjusted the language.
>
> > I did not find the discussion of the related works to be sufficient. The authors should expand that discussion so that the readers understand the differences and similarities with the proposed method better.
>
> We are working on it.
>
> > I find that the explanation of Wyckoff position in the third paragraph of the introduction is not easy to understand. It may be too early in the paper to go into such an explanation.
>
> Thank you for sharing this feedback, we have edited the introduction to hopefully improve clarity. Structuring the paper indeed presents a conundrum to which we haven't been able to find a perfect answer. Space group symmetry is a complex concept - that we are forced to explain within the page limit. And since it's necessary for understanding of WyFormer and related work, we have to put it early.
>
> > There are some typos and mistakes that the authors should look into correcting. For example, "lattice transition -> lattice translation" or " Cordiality -> Cardinality".
>
> > I don't see what the footnote 1 adds to the discussion, I find it more confusing than anything. Could the authors clarify it, or consider simply removing it?
>
> > I don't understand the expression in the abstract "These symmetries form energy configurations". What is meant there?
>
> Fixed, thank you.

---

> ### Author Response · Authors · 2024-11-26
> **Response to otui - enumerations and spherical harmonics**
>
> > The proposed representation for Wyckoff positions is universal across space groups but might not allow proper generalization since the "enumeration" variable is not grounded on physical information but on an arbitrary convention. Therefore, if a group is rare in the training data (this is indeed the case for datasets like MP20), there is no reason that the model will learn to capture that variable correctly. The authors should discuss this limitation appropriately.
>
> Indeed “enumeration” is unphysical and inelegant. When we switched from Wyckoff letters to site symmetries this last bit of the arbitrariness remained, and has been an eyesore ever since. After processing the comment we thought of a better representation. Every Wyckoff position is uniquely defined by a set of symmetry operations. For example, let us take space group 10 and Wyckoff positions 2k and 2l:
>
> ```jsx
> Wyckoff position 2k in space group 10 with site symmetry 2
> 0, y, 1/2
> 0, -y, 1/2
>
> Wyckoff position 2l in space group 10 with site symmetry 2
> 1/2, y, 1/2
> 1/2, -y, 1/2
> ```
>
> We apply those operations to points [0, 0, 0] and [1,1,1]:
>
> ```
> 2k:
> array([[[ 0. ,  0. ,  0.5],
>         [ 0. ,  1. ,  0.5]],
>        [[ 0. ,  0. ,  0.5],
>         [ 0. , -1. ,  0.5]]])
>  2l:
>  array([[[ 0.5,  0. ,  0.5],
>         [ 0.5,  1. ,  0.5]],
>
>        [[ 0.5,  0. ,  0.5],
>         [ 0.5, -1. ,  0.5]]])
> ```
>
> This way each operation is represented by two vectors, and each Wyckoff position is represented by 2 x number of operations it contains. The last remaining step is to get a fixed-length description of the Wyckoff position. We do this by convolving the points with spherical harmonics:
>
> $\text{WP} = \{\{x_1, y_1, z_1\}, …, \{x_k, y_k, z_k\}\}$, where k is 2 x the number of operations in the WP
>
> $\phi_i = \arctan(y_i, x_i)$
>
> $\theta_i = \arccos(z_i)$
>
> $H(\text{WP}) = \sum_{i=1}^k [Y_n^0 (\theta_i, \phi_i), …, Y_n^n (\theta_i, \phi_i)]$ resulting in a vector of size n. $n = 2$ is enough to disambiguate to disambiguate all Wyckoff positions with the same site symmetry belonging to the same space groups; $n=1$ is not. We have tried just averaging, without the harmonics, but it failed to disambiguate some WPs.
>
> What do you think of the idea? We have implemented it in code, will add some experiments.

---

> ### Author Response · Authors · 2024-11-28
> **Review response: otui - enumerations alternative implemented**
>
> Following your comment, we have devised and implemented the physics-grounded alternative to *enumerations,* it is now used for all property prediction results. The representation is described in Appendix B “Spherical harmonics”, and we now perform a small ablation study, Appendix M “Performance analysis of encoding WPs with spherical harmonics”.
>
> We edited the related work section to clearly place WyFormer in the ecosystem of material generative models, feedback is most welcome.

---

> > ### Comment · Reviewer_otui · 2024-11-28
> > **Response**
> >
> > I appreciate the authors efforts that have resulted in an improved paper. I will maintain my score since I still think some aspects of presentation and discussion could be polished and improved.

---

> ### Comment · Reviewer_otui · 2024-11-30
> **Response 2**
>
> I must say that I am also having difficulties tracking the updates on the paper. Could the authors color them in a different way?
>
> In particular, the authors have responded to my comment about generalization of the representation by introducing a method based on spherical harmonics. But in the paper a method based on a discrete representation of transformations (sec 2.2) is also mentioned. I noticed that a recent workshop paper [1] and ICLR submission introduced this representation. I guess the authors meant to cite that paper instead of the unrelated paper [2]? I find this very confusing.
>
> If yes, this has to be clarified and credit should be correctly attributed to [1]. In addition, even if the submission is concurrent, the authors should discuss this very related work, since they know about it (it is cited elsewhere in the paper) and they have used ideas from it.
>
> I also don't understand if the experimental results were updated using this discrete representation or the spherical harmonics based one.
>
> It is important that the authors address these points.
>
> [1] D. Levy, S. S. Panigrahi, S.-O. Kaba, Q. Zhu, M. Galkin, S. Miret, and S. Ravanbakhsh. Symmcd: Symmetry-preserving crystal generation with diffusion models. In AI for Accelerated Materials Design- NeurIPS 2024.
>
> [2] Y. Bengio, S. Lahlou, T. Deleu, E. J. Hu, M. Tiwari, and E. Bengio. Gflownet foundations. The Journal of Machine Learning Research, 24(1):10006–10060, 2023.

---

> > ### Author Response · Authors · 2024-12-02
> > **Response to otui**
> >
> > > I must say that I am also having difficulties tracking the updates on the paper. Could the authors color them in a different way?
> >
> > OpenReview provides the ability to compare revisions [here](https://openreview.net/revisions?id=ursX3k1rTO); we can no longer upload a new pdf until the camera-ready version, unfortunately.
> >
> > At the same time, we’ve been accumulating the changes for the camera-ready version [here](https://www.notion.so/Camera-ready-drafts-15075a35da3680728793d712979a9e75?pvs=21). We are not sure how appropriate it is to offer a changed pdf after the pdf modification deadline, so we in no way insist the reviewers open it. The changes are detailed in the comment below.
> >
> > > In particular, the authors have responded to my comment about generalization of the representation by introducing a method based on spherical harmonics. But in the paper a method based on a discrete representation of transformations (sec 2.2) is also mentioned. I noticed that a recent workshop paper [1] and ICLR submission introduced this representation. I guess the authors meant to cite that paper instead of the unrelated paper [2]? I find this very confusing.
> >
> > Apologies for the wrong citation. The intended citation was [Crystal-GFN](https://arxiv.org/pdf/2310.04925) from the 2023 workshop, by the same group as GFlowNet, hence the mistake.
> >
> > To clarify different representations used in WyFormer:
> >
> > 1. Space group of the whole crystal is represented as the one-hot encoded matrix of symmetry operations per axis from pyXtal + one-hot Bravais lattice type.
> > 2. WP site symmetry is represented as a categorical token, e. g. `m-3m`, with learnable embedding.
> > 3. *Enumeration* corresponds to enumerating the WPs with the same site symmetry
> > 4. Spherical harmonics are based on applying all the symmetry operations for each WP to points `[0,0,0]` and `[1,1,1]`; they do not use the one-hot encoded matrix of symmetry operations per axis. We have thought of it, but such representation will not allow disambiguation of WPs with the same site symmetry.
> >
> > Only part 4 changed during the revision process.
> >
> > > If yes, this has to be clarified and credit should be correctly attributed to [1]. In addition, even if the submission is concurrent, the authors should discuss this very related work, since they know about it (it is cited elsewhere in the paper) and they have used ideas from it.
> >
> > We have carefully read [1] and outline similarities and differences:
> >
> > - Similarities
> >     - Both models generate crystals conditioned on space group symmetry
> >     - Both models use representations based on Wyckoff positions, with differences listed below
> >     - Both models use the same representation of the space groups
> >     - Both models use WP site symmetries as opposed to Wyckoff letters
> > - Differences
> >     - [1] is a diffusion model, ours is an autoregressive
> >     - To model probabilities [1] uses a message-passing GNN, we use a Transformer encoder
> >     - [1] also predicts crystal lattice and fractional coordinates
> >     - WP site symmetry representation is different: [1] uses the same one-hot encoded symmetry per axis as for the space group symmetry, we use categorical variables with learnable embeddings
> >     - Handling of WPs with the same site symmetry. Since [1] also generates fractional coordinates, they rely on them to disambiguate WPs with the same site symmetry. We use *enumerations* or spherical harmonics.
> >
> > We will be happy to add this analysis to the camera-ready version.
> >
> > Interestingly enough, in terms of evaluation, [1] does not publish the novelty rate of the generated structures, only template novelty. Judging by the ratio of the S.U.N. rate to stability rate in table 3, it should be ~74%.
> >
> > In terms of allocation of credit. We arrived at this space group representation independently, first from from crystallography, then after learning about Crystal-GFN and discovering that pyXtal provides a convenient function `to_matrix_representation_spg()`. At the same time, upon further investigation, this function was [committed](https://github.com/MaterSim/PyXtal/commit/03079c5c28743ad57fa194280666d6ed8531eaf1) by one of the authors of [1], so we err on the side of recognition, and gladly credit them. We’ve also credited them template novelty since their workshop submission (although not its publication) predates our ICLR submission.
> >
> > > I also don't understand if the experimental results were updated using this discrete representation or the spherical harmonics based one.
> >
> > Property prediction results use spherical harmonics; generative results use enumerations, the generated structure sample hasn’t changed since the paper submission. This is specified in section 2.2, paragraphs “De novo generation” and “Property prediction”. Generation with the spherical harmonics is possible, we have implemented and are in the process of evaluating the method described in Appendix N “Sampling harmonic — encoded WPs”

---

### Official Review · Reviewer_7MTL · 2024-11-02

**Soundness:** 3
**Presentation:** 2
**Contribution:** 3
**Rating:** 5
**Confidence:** 3

**Summary:**

This paper proposes a transformer-based approach that leverages Wyckoff positions to encode material symmetries efficiently. This is done by primarily encoding the discrete symmetries of space groups without using atomic coordinates. The discussion on WyFormer, including tokenization and (extensive) metrics, is detailed. Their main contribution is to represent a crystal as an unordered set of tokens and make de novo material and property predictions. Furthermore, four new metrics are proposed (P1, Template Novelty, Space Group, and S.S.U.N.) to judge the ability to reproduce symmetry properties accurately. Results indicate that WyFormer outperforms other methods in terms of template novelty, space group distribution, and fraction of asymmetric structures.

Overall I think the paper could be a good step in the direction of using symmetries for property prediction, provided certain clarifications on experimental details and broader evaluations are addressed.

**Strengths:**

- Crystal representation for tokenization.
- Material property prediction results are surprisingly good when compared against neural nets trained for energy prediction.
- Four new metrics provide a new way of looking at models' ability to generate symmetry properties.
- Justification in the appendix for the selected two structure generation methods.

**Weaknesses:**

- The scope of material property prediction- authors focus on just two (energy and band gap). If feasible, can the authors provide some insight on which other properties could be predicted, purely from a correlation with crystal structure perspective?
- The proposed method is evaluated on a single dataset, MP-20, and makes it hard to judge the generalizable nature of WyFormer from it. Are there other datasets on which performance can be evaluated?

**Questions:**

1. In section 1.3, line 138, "...our main differences are listed in the discussion of our contributions 1.2."  where are the main differences listed in section 1.2? Or am I missing something?

2. Can the authors explain why the Space Group value for WyForDiffCSP++ is high while the S.S.U.N. value is similar to WyCryst in Table 1a?

3. A discussion on computational cost would be good to have, given that the authors mention that the entire dataset fits into GPU memory (training time and memory requirements)

4. Are there methods apart from CHGNet that improve crystal structure generation?

5. Have the authors tried other property prediction experiments besides energy and band gap?

Additional Feedback:
1. line 280: "they to be" -> "they are"?
2. line 282: percetage -> percentage?

---

> ### Author Response · Authors · 2024-11-12
> **Quick question**
>
> Thank you very much for the detailed and to-the-point review! We will address the issues in the upcoming two weeks. Could you please elaborate, what did you mean in
> > Are there methods apart from CHGNet that improve crystal structure generation?
>
> Whether it's possible to use another method in place of CHGNet?

---

> > ### Comment · Reviewer_7MTL · 2024-11-12
> > **Response to quick question**
> >
> > Yes, are there other methods apart from CHGNet that could be used?

---

> > > ### Author Response · Authors · 2024-11-26
> > > **CHGNet alternatives**
> > >
> > > Yes, of course, any machine learning interatomic potential. Wyckoff Transformer is future-proof in this regard. CHGNet paper was released a year ago, the latest and the greatest potential now is [eqV2 published in November 2024](https://github.com/FAIR-Chem/fairchem).

---

> > > > ### Comment · Reviewer_7MTL · 2024-11-28
> > > > **Official Comment by Reviewer 7MTL**
> > > >
> > > > I acknowledge the rebuttal and thank the authors for it. Considering all the comments that have been made, I will be staying with my score.

---

> > > > > ### Author Response · Authors · 2024-11-30
> > > > > **Response to 7MTL: training time**
> > > > >
> > > > > We are pleased to present the full report on training time and memory requirements under [this anonymized URL](https://wyckoff.notion.site/Training-generation-computational-requirements-14e75a35da36806cad56f3f955530779), we will also add it to the camera-ready version. In short, WyFormer training time is 11h to DiffCSP++ 19.5h and GPU memory required is an order of magnitude lower: 2000 to 32000 MiB.

---

> ### Author Response · Authors · 2024-11-26
> **Response to 7MTL pt 1**
>
> > The scope of material property prediction- authors focus on just two (energy and band gap). If feasible, can the authors provide some insight on which other properties could be predicted, purely from a correlation with crystal structure perspective?
>
> We are running experiments on the [AFLOW dataset](https://www.nature.com/articles/s41524-021-00545-1) containing data on electrical and mechanical properties such as bulk, shear, conductivity, etc. for about five thousand crystals. The results will be included in the updated version of the pdf. In addition, we plan to try to use our model to predict the critical temperature of superconductivity; these results will be added if we manage to complete the necessary experiments.
>
> > The proposed method is evaluated on a single dataset, MP-20, and makes it hard to judge the generalizable nature of WyFormer from it. Are there other datasets on which performance can be evaluated?
>
> MP-20 is a highly representative dataset, it contains almost all experimentally verified materials within 0.08 eV/atom of convex hull in the Materials Project database from around July 2021 - a sizable proportion of all materials known to humanity. It's not obvious how to generalise more, there is only one physical world after all. That being said, we are running on MPTS-52, a subset of Materials Project with the increased limit on the number of sites from 20 to 52. We are not aware of any published results for de novo generation on this dataset though, so the question of the results interpretation will arise, any suggestions will be most welcome. There are also two artificial datasets: [perov-5](https://github.com/txie-93/cdvae/tree/main/data/perov_5) and [carbon-24](https://github.com/txie-93/cdvae/tree/main/data/carbon_24). They lack in diversity, and do not ensure the stability of the structures in them, limiting the utility of conclusions drawn from them; Xie et al who introduced them for generative modelling explicitly [write](https://github.com/txie-93/cdvae/tree/main/data/perov_5) about perov-5 that a significant portion of the materials are not thermodynamically stable, i.e., they will decompose to nearby phases and cannot be synthesized. Recent works such as [FlowMM](https://arxiv.org/pdf/2406.04713) and [MatterGen](https://arxiv.org/pdf/2312.03687) no longer use those two datasets for benchmarking de novo generation.
>
> > In section 1.3, line 138, "...our main differences are listed in the discussion of our contributions 1.2." where are the main differences listed in section 1.2? Or am I missing something?
>
> We have expanded the related work discussion
>
> > Can the authors explain why the Space Group value for WyForDiffCSP++ is high while the S.S.U.N. value is similar to WyCryst in Table 1a?
>
> This is very insightful question, thank you. *Space Group* reflects the similarity of the distribution of the space groups. WyFormer, and, by extension WyForDiffCSP++ is explicitly conditioned on space group number, ensuring that it closely matches the distribution of the space groups in the dataset. WyCryst, on the other hand, samples latent space near the training examples, resulting in a more approximate reconstruction of the space group distribution. Moreover, Table 1a values are computed using novel structures only - and WyCryst has novelty of just 52% to 89% for WyForDiffCSP++ distorting the distribution further. See Figure 1 in the updated pdf for an illustration.
>
> *S.S.U.N.* does not rely on the whole space group distribution, it simply verifies whether any higher symmetry group present, the opposite of column *P1 (%)* in the table, where WyForDiffCSP++ has 1.4%, and WyCryst 4.8%  - a worse, but still comparable value.
>
> > A discussion on computational cost would be good to have, given that the authors mention that the entire dataset fits into GPU memory (training time and memory requirements)
>
> We will add a corresponding section to the paper when the latest experiments finish. In short, training on GPU in generative mode takes ~11 hours and property prediction ~1 hour for mp-20 dataset
>
> > Have the authors tried other property prediction experiments besides energy and band gap?
>
> We are running experiments on the [AFLOW dataset](https://www.nature.com/articles/s41524-021-00545-1) containing data on electrical and mechanical properties such as bulk, shear, conductivity, etc. for about five thousand crystals. The results will be included in the updated version of the pdf. In addition, we plan to try to use our model to predict the critical temperature of superconductivity; these results will be added if we manage to complete the necessary experiments.
>
> Thank you for the text comments, we have fixed them.

---

> ### Author Response · Authors · 2024-12-03
> **Response to 7MTL - an additional dataset**
>
> We are happy to present an evaluation of WyFormer on the more challenging MPTS-52 dataset at [this anonymized url](https://wyckoff.notion.site/MPTS-52-15175a35da36802c9946c278c6f12f28), to be included in the camera-ready version. In short, WyFormer can de novo generate stable novel realistic structures with up to 52 atoms per unit cell, it is the only generative model that demonstrated such capability we are aware of.
>
> With this, we have addressed the last remaining concern. It's less than a day until the deadline, but if you find an opportunity to give  feedback, in particular on the results for de novo generation on MPTS-52 and property prediction on AFLOW (Section 3.2), we would be extremely grateful.

---

### Official Review · Reviewer_DeYQ · 2024-11-04

**Soundness:** 2
**Presentation:** 3
**Contribution:** 3
**Rating:** 5
**Confidence:** 3

**Summary:**

The paper focuses on the tasks of de novo materials generation and materials property prediction. The main contribution is a Wyckoff representation tokenization and model training strategy. For de novo generation, once the transformer generates a Wyckoff position then PyXtal and CHGNet are used to generate/relax the structure. In particular, the model is good at generating materials with the proper space group.

**Strengths:**

- The application of ML to materials discovery is interesting and timely.
- The Wyckoff representation builds in crystal symmetries in a natural way.
- Good empirical results for template novelty, P1, and Space Group metrics.
- The paper is written well

**Weaknesses:**

- Little improvement in standard de novo generation metrics. SUN actually goes down compared to DiffCSP.
- Additionally, de novo generation metrics were computed with a ML potential instead of DFT.
- The property prediction benchmark is not particularly compelling because there are better benchmarks out there with other more recent models as baselines (e.g. CHGNet), such as Matbench discovery.
- If not there, it would be good to include this citation (https://arxiv.org/abs/2106.11132).

**Questions:**

- You found a nice way to tokenize a Wyckoff representation, would it be better to fine-tune a LLM with this representation than train a transformer from scratch? The CrystalLLM paper (https://arxiv.org/abs/2402.04379v1) had some nice results that could potentially be improved with your representation.
- Another important axis is the inference speed or cost of de novo generation, how does WyFormer or WyForDiffCSP++ compare to DiffSCP/FlowMM?
- Is there a way to more concretely show the benefit template novelty?
- Can you plot the distribution of space groups generated from WyFormer compared to MP-20 distribution?

---

> ### Author Response · Authors · 2024-11-26
> **Response to DeYQ**
>
> 1. We have computed some DFT, and updated the results accordingly. The overall conclusion is the same - WyFormer still outperforms the other models conditioned on space group, DiffCSP still has an advantage in stability.
> 2. We have added CHGNet to the table. As you correctly note, in terms energy prediction our generative model indeed is surpassed by dedicated forward models — already just on Materials Project data. At the same time, all the models, including WyFormer, perform within the accuracy of DFT itself, which is the main conclusion we want to draw. We are open to running on matbench-discovery, but not sure whether we can manage before the discussion deadline. The current work is focused on a generative model, building and tuning a proper forward model based on WyFormer representation is ongoing work and we envision this will be a follow-up paper.
> 3. Cited https://arxiv.org/abs/2106.11132
> 4. We have performed experiments with fine-tuning an LLM, reported in the "Fine-tuning LLM with Wyckoff representation" section in Appendix. In short, LLM performs similarly according to the proxy metrics. And it is orders of magnitude more computationally expensive - speaking of inference time.
> 5. Added space groups distribution to the paper as Figure 1.
> 6. We are working on template novelty.

---

> ### Author Response · Authors · 2024-11-26
> **Response to DeYQ - inference speed**
>
> We conducted experiments on a machine with NVIDIA RTX 6000 Ada and 24 physical CPU cores
>
> **WyFormer**
>
> Generating a batch of $10^5$ Wyckoff representations takes 25 seconds, of which 5 seconds are spent generating pytorch tensors, and 20 seconds on decoding them into Python dictionaries containing Wyckoff representations. In total, Wyckoff representation generation takes 0.05 GPU milliseconds and 4.8 CPU milliseconds per structure.
>
> Obtaining unrelaxed structures using pyXtal takes 100 CPU core milliseconds / structure.
>
> Finally, relaxing the structure is the most expensive step:
>
> 1. DiffCSP++ takes 14 minutes to produce 1000 structures at 840 GPU milliseconds / structure. Note that we modified the code to remove the inference of atom types, so it runs faster compared to the original version.
> 2. CHGNet: 112 GPU seconds / structure for MP-20 on NVIDIA A40
>
> **Baselines**
>
> DiffCSP: the authors don’t report speed. On our machine, generating 10000 structures on GPU took 1 hour, at 360 GPU milliseconds per structure.
>
> DiffCSP++: the authors don’t report speed. On our machine, generating 27135 structures took 6 hours, at 1.25 GPU seconds per structure
>
> CrystalFormer paper: “It takes 520 seconds to generate a batch size 13,000 crystal samples on a single A100 GPU, which translates to a generation speed of 40 milliseconds per sample.”
>
> FlowMM: The authors also do not publish inference time or model weights. They claim to be 3x faster than DiffCSP in terms of integration steps, which are not defined for WyFormer.
>
> WyCrst paper: “Latent space sampling 1 CPU second/2000 structures; PyXtal generation 2 CPU core seconds/structure”
>
> **Conclusion**
>
> WyFormer model by itself is 3 orders of magnitude faster than DiffCSP/FlowMM, but the main question when analysing inference speed is what are the requirements of the next analysis step?
>
> There are three common options:
>
> 1. A classic high-throughput material discovery pipeline with DFT computations for all the candidate materials, which is at least three orders of magnitude slower (~110 CPU hours/structure for MP-20 type data) compared to all the generative models, rendering the generative model inference cost insignificant.
> 2. Using a machine learning interatomic potential as a replacement for DFT. Again, this is at least ~100 times slower than the generative models that prepare inputs for it, with WyFormer being the fastest of them all.
> 3. Using the generated structures as they are as an input for a property prediction model. The biggest question is the reliability of such set up, as raw generated structures will be out of domain for the predictor. In this case WyFormer + DiffCSP++ works slightly faster compared to just DiffCSP++. If the predictive model accepts Wyckoff representations, for example [Wren](https://arxiv.org/abs/2106.11132) which you mention in the comment, WyFormer becomes at least 100 times faster compared to alternatives.

---

> ### Comment · Reviewer_DeYQ · 2024-11-27
>
> Thanks for your response. The DFT results and space group distributions are really nice additions. I still do not find the property prediction aspect of paper compelling, if it is not central to paper it might be worth moving to the appendix. Also, I find it questionable that CHGNet would perform nearly the same as SchNet??
>
> Given your response it seems there is a case to be made for WyFormer in terms of the computational cost for generating SUN/SSUN structures. The code for many of the models in your comparison including FlowMM is public so the inference cost could be compared, and the downstream costs of DFT or an MLIP can be considered fixed. While I believe these results could be interesting they are not formalized so I am staying with my score.

---

> ### Author Response · Authors · 2024-11-27
> **Response to DeYQ - properties and inference time**
>
> > I find it questionable that CHGNet would perform nearly the same as SchNet??
>
> Indeed. We are double-checking the references.
>
> > Given your response it seems there is a case to be made for WyFormer in terms of the computational cost for generating SUN/SSUN structures. The code for many of the models in your comparison including FlowMM is public so the inference cost could be compared, and the downstream costs of DFT or an MLIP can be considered fixed. While I believe these results could be interesting they are not formalized so I am staying with my score.
>
> We have added a table with the formalised report of inference times to Appendix D "Inference speed". We show formally that keeping the downstream costs of DFT/MLIP fixed, WyFormer is 4 orders of magnitude less GPU intensive than baselines, and 3 times less CPU. The better performance is achieved by it's autoregressive sampling as opposed to iterative improvement by diffusion models. CPU costs are dominated by pyXtal, in case the structures are used as inputs to a Wyckoff-based property prediction, they can be avoided.

---

> ### Author Response · Authors · 2024-11-28
> **Responce to DeYQ - template novelty and property benchmarks**
>
> > Is there a way to more concretely show the benefit template novelty?
>
> In materials science, template novelty is the possibility of atoms that constitute the compound to arrange in a completely new symmetry, thereby giving it a new physical property and hence this is extremely beneficial. For instance, [Hicks et al. (AFLOW)](https://www.nature.com/articles/s41524-020-00483-4) use Wyckoff positions as a part of their prototype label; A well-founded criticism of GNoME by [Leeman et. al.](https://journals.aps.org/prxenergy/pdf/10.1103/PRXEnergy.3.011002) shows that no new materials have in fact been discovered in the select sample of 43 put forward by GNoME team after [doing more than 200k DFT computations](https://www.nature.com/articles/s41586-023-06735-9). If GNoME was designed as a generative model capable of producing new templates, as opposed to template-preserving element substitution, this could have been avoided.
>
> We are happy to report additional analysis to support our ability to generate new templates. Lack of template novelty limits sample diversity which in turn very concretely limits the total number of unique structures that the model is capable of generating. To concretely show our advantage in template novelty, we have now additionally sampled 118k examples from DiffCSP++ and WyFormer. DiffCSP++ sample is 67% unique, whereas WyFormer sample is 90% unique. The results are included in Appendix I. "Template Novelty and Diversity”. N. B. The sample size is limited by the great computational cost of DiffCSP++
>
> > I find it questionable that CHGNet would perform nearly the same as SchNet??
>
> 33 meV SchNet MAE was reported by [Lin et al](https://proceedings.mlr.press/v202/lin23m.html); 30 meV CHGNet MAE was reported [Deng et al.](https://www.nature.com/articles/s42256-023-00716-3) There was a discrepancy: the reported value for SchNet was for formation energy, while for CHGNet it was for the total potential energy. To address this discrepancy, we computed formation energy using CHGNet on the MP-20 dataset, the result is similar: 34 meV. We would like to again raise the point we make in the paper: the error of DFT computation of formation energy is around 80 meV ([Jha et al.](https://www.nature.com/articles/s41467-019-13297-w)), it is a very reasonable outcome that trying to train a machine learning method beyond this accuracy leads to diminishing returns.
>
> Additionally, as a requested by the other reviewers, we are happy to invite you to consider WyFormer’s competitive performance on the AFLOW dataset; we find it especially satisfying that WyFormer shows the best performance in predicting thermal conductivity, as its dependence on the space group symmetry follows from the first order approximation of kinetic theory, where higher symmetry crystals typically have higher thermal conductivity due to higher group velocities and longer scattering times (and hence mean free paths) due to lower anharmonicity [Newnham (2004)](https://doi.org/10.1093/oso/9780198520757.003.0020); [Yang et al. (2021)](https://link.aps.org/doi/10.1103/PhysRevB.103.184302). Thus, symmetry-reliant properties are better predicted by our symmetry-aware generation model.
>
> While competitive in each metric individually, the reason WyFormer beats other contemporary models lies in the synergy of different aspects:
>
> 1. Quality and highest diversity of the generated structures
> 2. Quality of property prediction
> 3. Inference speed
>
> With the generation cost of 0.05 GPU ms per structure and property prediction ability, WyFormer is ideal for running a large-scale property screening campaign. For example, we are currently working on evaluating whether P1 structures produced by DiffCSP and FlowMM truly lie on the hull - and this requires screening a large number of hull candidates; but it is beyond the scope of our current study.

---

> ### Author Response · Authors · 2024-12-02
> **Request for final feedback from DeYQ**
>
> Dear DeYQ, are are thankful for your comments that led to a host of additional evaluations strengthening our approach:
> 1. Using Wyckoff representation with a fine-tuned LLM in Appendix L, which leads to similar values of proxy metrics, at a much greater computational cost.
> 2. Making property prediction results much more compelling by showing competitive performance on a new dataset with 4 new properties, adding CHGNet to energy baselines in Section 3.2
> 3. Showing superb computational performance in Appendix D
> 4. Showing a concrete benefit of template novelty as an increase in the ability of a model to produce large number of unique structures in Appendix I
>
> We are eagerly awaiting your feedback; the reviewer response deadline, midnight December 2nd AoE is just ~19.5 hours away.

---

### Author Response · Authors · 2024-12-04
**Closing argument**

We are deeply thankful to the reviewers for their time and effort that made WyFormer a better model with an even larger number of experiments to prove it. Here is our closing argument that summarizes the discussion and makes the final case for our paper.

**Strengths**
1. Application of ML to materials discovery is an important goal [DeYQ, hhjg]
2. Good representation:
    1. The Wyckoff representation builds in crystal symmetries in a natural way. [DeYQ]
    2. Strengths: Crystal representation for tokenization. [7MTL]
    3. Originality: The Wyckoff Transformer introduces a novel approach to crystal generation by utilizing Wyckoff positions to encode symmetries explicitly, making it unique among generative models … a creative and effective innovation for materials science. [hhjg]
3. Introduction and motivation of symmetry-based metrics and good performance according to them:
    1. Good empirical results for template novelty, P1, and Space Group metrics. [DeYQ]
    2. Four new metrics provide a new way of looking at models' ability to generate symmetry properties. [7MTL]
    3. The evaluation metrics for symmetry and novelty of structures based on Wyckoff templates is also valuable. [otui]
    4. The evaluation methods, including the newly proposed methods for evaluating symmetry, form a compelling discussion section. The method demonstrates effective gains for the symmetry metrics and is competitive for widely-used proxy metrics. [rFbw]
    5. The paper includes rigorous experimental results, showing the model’s success in generating symmetric crystal structures … The evaluation is thorough, comparing the model’s performance to state-of-the-art methods across multiple metrics, demonstrating its robustness and effectiveness in real-world scenarios. [hhjg]
4. Good property prediction performance:
    1. Material property prediction results are surprisingly good when compared against neural nets trained for energy prediction. [7MTL]
5. Low inference computational resources requirements:
    1. Another important axis is the inference speed or cost of de novo generation … it seems there is a case to be made for WyFormer in terms of the computational cost for generating SUN/SSUN structures. [[DeYQ](forum?id=ursX3k1rTO&noteId=Kl8xL7aUbA)]
6. The proposed solution is simple and sound [otui]

**Neutral**

1. Performance according to the standard metrics. DeYQ considers the lack of gain to be a weakness, while rFbw cosiders competitive performance to be a strength
2. Training resources requirements (7MTL, rFbw). WyFormer takes similar time to train compared to diffusion-based baselines, but an order of magnitude less GPU memory [[anonymized url](https://wyckoff.notion.site/c)].

**Weaknesses**

1. Evaluation on a single dataset (7MTL, rFbw). We address it in three ways:
    1. Conceptual. We have only one physical world, and the goal of the model is to generate novel stable materials in this world; MP-20 is a diverse dataset and contains a sizable fraction of all known materials.
    2. We show WyFormer generative performance on MPTS-52, a more challenging dataset containing materials with up to 52 atoms per unit cell; it is the only generative model that demonstrated the capability to handle it that we are aware of. [[anonymized url](https://wyckoff.notion.site/mpts)]
    3. We show WyFormer’s competitive performance in property prediction on AFLOW dataset (Table 5)
2. Data fragmentation because of conditioning on symmetry space group (rFbw, otui). We address this by using transferable space group and Wyckoff position representations. In the initial version of the paper one part of the representation was not transferable between space groups, during the review period we have developed a transferable representation for this part as well (Appendix B).
3. Use of CHGNet for stability evaluation (DeYQ). We have run DFT evaluations that, despite the limited sample size, show competitive performance of WyFormer according to stability metrics, and positive correlation between CHGNet and DFT estimations of stability (Table 1)
4. Only predicting energy and band gap (7MTL, DeYQ). We added experiments showing competitive performance of WyFormer in predicting thermal conductivity, Debye temperature, bulk modulus, and shear modulus on AFLOW dataset (Table 5)
5. Related work discussion (otui, hhjg). We have expanded, clarified and systematized the relevant work section.
6. Presentation (otui, rFbw, hhjg); DeYQ, however, commended the writing. We have extensively edited, clarified, and expanded the writing, added and improved illustrations.

In summary, WyFormer has a unique and powerful synergy of attributes, proven by extensive experimentation: best-in-class symmetry-conditioned generation, physics-motivated inductive bias, competitive stability of the generated structures, competitive property prediction quality, and unparalleled inference speed. Altogether, they make it a great starting point in the material design pipeline.

---

### Meta-Review · Area_Chair_GAHr · 2024-12-22

**Metareview:**

The authors propose Wyckoff Transformer, a generative model for material crystals that obey space group symmetry. The key algorithmic idea is that transformers can provide a permutation-invariant, autoregressively-generated, discrete representation of inorganic material structures. Using this representation the authors show performance of the model on de novo material generation along with several property prediction tasks.

The method proposed in the paper is intuitive, and addresses an important problem in de novo crystal generation with current AI models (ie, the lack of symmetry). However, issues were raised regarding the clarity of the exposition; the strength of the results (both the de novo generation and the property prediction tasks); potential limitations with the representation that may not have been properly highlighted in the narrative; and conceptual overlap/attribution of previous work.

This was a close call, but I hope that the authors will take into account the extensive feedback provided by the reviews while preparing revisions.

**Additional Comments On Reviewer Discussion:**

This paper was reviewed by several experts in the field. Initial reviews were somewhat on the negative side, but the authors responded  with several additional experimental results, and clarifications in the writing --- to the extent that the paper was significantly re-written. In the end, the following big issues remain that prevent publication in my opinion:
- Writing clarity: I echo the reviewers in this aspect; from my personal reading the paper still is in need of major revisions, particularly in the introduction, for an average ICLR audience that may not have the requisite chemistry background.
- Strength of results: the method does not show very clear gains particularly in de novo generation, and does not compare on existing benchmarks (such as matbench) for property prediction.

---

### Decision · Program_Chairs · 2025-01-22

Reject